EMBO
Molecular Medicine

# Inhibition of Drp1/Fis1 interaction slows progression of amyotrophic lateral sclerosis

Amit U Joshi[1] (ID), Nay L Saw[2], Hannes Vogel[3], Anna D Cunnigham[1], Mehrdad Shamloo[2] & Daria Mochly-Rosen[1,*] (ID)

## Abstract

Bioenergetic failure and oxidative stress are common pathological hallmarks of amyotrophic lateral sclerosis (ALS), but whether these could be targeted effectively for novel therapeutic intervention needs to be determined. One of the reported contributors to ALS pathology is mitochondrial dysfunction associated with excessive mitochondrial fission and fragmentation, which is predominantly mediated by Drp1 hyperactivation. Here, we determined whether inhibition of excessive fission by inhibiting Drp1/Fis1 interaction affects disease progression. We observed mitochondrial excessive fragmentation and dysfunction in several familial forms of ALS patient-derived fibroblasts as well as in cultured motor neurons expressing SOD1 mutant. In both cell models, inhibition of Drp1/Fis1 interaction by a selective peptide inhibitor, P110, led to a significant reduction in reactive oxygen species levels, and to improvement in mitochondrial structure and functions. Sustained treatment of mice expressing G93A SOD1 mutation with P110, beginning at the onset of disease symptoms at day 90, produced an improvement in motor performance and survival, suggesting that Drp1 hyperactivation may be an attractive target in the treatment of ALS patients.

**Keywords** amyotrophic lateral sclerosis; dynamin-related protein 1; fission 1; mitochondrial dysfunction; Protein–Protein interactions
**Subject Categories** Genetics, Gene Therapy & Genetic Disease; Neuroscience

## Introduction

Amyotrophic lateral sclerosis (ALS), which clinically manifests by progressive muscle atrophy and paralysis, is a fatal neurodegenerative disease characterized by the death of upper and lower motor neurons (MN; Boillee *et al*, 2006). Individuals with ALS most commonly die of respiratory failure or pneumonia within 3–5 years from initial diagnosis (Pisa *et al*, 2016). Currently, the glutamate release inhibitor, riluzole, and recently approved free radical scavenger, edaravone, are the only medications approved by the FDA for ALS, and therefore, there remains a strong need for new treatment strategies (Lacomblez *et al*, 1996; Faes & Callewaert, 2011; Ittner *et al*, 2015; Hardiman & van den Berg, 2017). Furthermore, eligibility for edaravone was restricted to patients with a relatively short disease duration and preserved vital capacity, indicating a need for a more encompassing treatment (Hardiman & van den Berg, 2017; Maragakis, 2017; Sawada, 2017). For most patients, the underlying cause for ALS is not known (sporadic ALS), but over 100 different mutations in superoxide dismutase 1 (SOD1) account for < 20% of familial ALS forms (Cozzolino & Carri, 2012). Transactive response (TAR) DNA-binding protein 43 (TDP-43) and fused in sarcoma (FUS) have also been genetically and pathologically linked to ALS; however, the underlying mechanisms by which these induce ALS pathology and the causal relationship between these events and the death of the motor neurons remain unclear (Mackenzie *et al*, 2010).

Several studies suggest possible defects in mitochondrial dynamics in models of ALS, regardless of the causative mutation (Menzies *et al*, 2002; Ehinger *et al*, 2015; Tafuri *et al*, 2015; Sharma *et al*, 2016). SOD1 resides in the mitochondrial intermembrane space, and abnormal mitochondrial morphology and cristae ultrastructure have been observed in mutant SOD1 mice and in ALS patient samples, predominantly in the spinal cord (De Vos *et al*, 2007; Song, Song *et al*, 2013). Mutant SOD1G93A affects mitochondrial dynamics, resulting in a significant decrease in mitochondrial length and an accumulation of round fragmented mitochondria (Tafuri *et al*, 2015). Abnormal mitochondrial dynamics was also recently observed in skeletal muscle of the SOD1 G93A mice (Luo *et al*, 2013), together indicating the importance of mitochondrial dynamics in ALS.

Mitochondria exist in the cells as highly dynamic entities, ranging from elaborate tubular networks to small organelles, through rapid and opposing processes of fission and fusion. Mitochondrial fission, the focus of our study, is mediated by the recruitment of Drp1, a cytosolic large GTPase, to the outer mitochondrial membrane by mitochondrial fission factor (Mff), mitochondrial dynamics proteins of 49 kDa and 51 kDa (Mid49/Mid51) and through fission 1 (Fis1; Loson *et al*, 2013; Osellame *et al*, 2016).

1 Department of Chemical and Systems Biology, Stanford University School of Medicine, Stanford, CA, USA
2 Behavioral and Functional Neuroscience Laboratory, Department of Neurosurgery, Stanford University School of Medicine, Stanford, CA, USA
3 Department of Pathology, Stanford University School of Medicine, Stanford, CA, USA
*Corresponding author. Tel: +1 650 724 8098; Fax: +1 650 723 4686; E-mail: mochly@stanford.edu

Whereas physiological fission is essential for maintaining mitochondrial quality (Shirihai et al, 2015), excessive Drp1-mediated fission causes mitochondrial fragmentation, mitochondrial membrane depolarization, increase in reactive oxygen production (ROS) and oxidative stress, and a decrease in ATP production and in other mitochondrial functions (Wu et al, 2011; Youle & van der Bliek, 2012; Babbar & Sheikh, 2013). Indeed, we found that inhibition of excessive Drp1 activity through blocking its interaction with Fis1 is protective in models of Parkinson's disease and Huntington's disease (Guo et al, 2013; Qi et al, 2013). Here, we determined whether Drp1 hyperactivation plays a role in the pathogenesis of ALS and whether inhibition of Drp1 hyperactivation through its interaction with Fis1 can reduce ALS pathology.

# Results

### ALS patient-derived fibroblasts show altered mitochondrial function and mitochondrial fragmentation associated with Drp1 hyperactivation

We first determined whether mitochondrial dysfunction is evident in fibroblasts of ALS patients carrying pathogenic mutations in SOD1 (I113T), in FUS1 (fused in sarcoma; R521G) or in TDP43 (TAR DNA-binding protein 43; G289S) genes. As fibroblasts have a mainly glycolytic metabolism, they were cultured in galactose-containing medium for 48 h, to induce dependence on oxidative phosphorylation (OX-PHOS) for ATP production (Aguer et al, 2011). In fibroblasts derived from ALS patients, the mitochondrial network was fragmented as compared with fibroblasts from healthy subjects (control), with prevalence of round-shaped mitochondria or sphere-like clusters (Fig 1A). To quantify the mitochondrial structure change, we utilized automated image analysis and examined the effects of these ALS mutations on mitochondrial morphology. ALS patient-derived fibroblasts carrying any one of these three mutations showed a ~50% decrease in mitochondrial interconnectivity score (1.01 vs. 0.48 for control and ALS, respectively; $P < 0.0001$) and elongation scores (1.54 vs. 0.77 for control and ALS, respectively; $P < 0.0001$) (Fig 1B and C). We next determined if this mitochondrial fragmentation was mediated by Drp1/Fis1 interaction, using P110, a heptapeptide conjugated to $TAT_{47-57}$ (TAT, for intracellular delivery) that selectively inhibits the interaction between Drp1 and Fis1, one of its adaptor proteins on the mitochondria (Qi et al, 2013). Treatment with P110 (1 μM/day for 2 days) significantly improved mitochondrial structure and improved the interconnectivity (from 0.48 to 1.32 following P110 treatment; $P < 0.0001$) (Fig 1A–C). Furthermore, to confirm that Fis1 is critical for the mitochondrial structural changes observed in ALS patient-derived fibroblasts, we transiently knocked down the expression Fis1 (Fig EV1A) and observed a significant recovery in mitochondrial structure as measured by previously described methods (Figs 1A and EV1B and C).

Superoxide dismutase 1 mutation caused a ~50% decrease in mitochondrial membrane potential (MMP; measured by TMRM) and in ATP production as compared to controls cells ($P < 0.0001$; Fig 1D and E). This mitochondria dysfunction was associated with increased oxidative stress; mitochondrial reactive oxygen species (ROS) and total ROS levels were 186% and 250% of control,

respectively, ($P < 0.0001$) in ALS patient-derived cells relative to control cells (Fig 1F and G). These mitochondrial defects were significantly reduced by treatment with P110 (1 μM/day for 2 days; Fig 1D–G).

Under physiological conditions, minimal levels of Drp1 are associated with mitochondria to maintain physiological mitochondrial fission. Drp1 recruitment from the cytosol to the mitochondrial outer membrane, a hallmark of activated mitochondrial fission (Frank et al, 2001), was ~threefold higher in ALS patient-derived fibroblasts relative to control cells, indicating Drp1 hyperactivation; this was significantly reduced by P110 treatment, to 1.9-fold relative to control cells ($P = 0.001$; Figs 1H and EV1D). More importantly, we assessed the specific interaction between Drp1 and Fis1 and observed that in ALS patient-derived fibroblasts, and there was a significant increase in their association, which was reduced by P110 treatment (Fig 1I and EV1G). We did not observe any significant changes in the total protein levels of either Drp1 or Fis1 proteins in these cells (Fig EV1F).

Previously, p62 (also known as SQSTM1), a protein implicated in protein aggregate formation and stalled autophagy, was shown to accumulate as the disease progresses in the G93A mouse spinal cord (Gal et al, 2007). Since p62 recruitment and accumulation at mitochondria have been associated with increased ROS production as well as mitochondrial membrane depolarization (Narendra et al, 2010), we next assessed the levels of p62 in the mitochondrial fraction in these patient-derived cells. Indicative of aberrant autophagy stall, we observed a 3.3-fold increased mitochondrial recruitment of p62, which was reduced by P110 treatment to 1.9-fold relative to controls (Fig EV1D and E), indicting an interplay between mitochondrial fragmentation via Drp1/Fis1 interaction and autophagy balance.

### NSC-34 cells expressing mutant SOD1 G93A exhibit mitochondrial dysfunction

To further dissect the pathways involved in P110-induced benefits observed in ALS patient-derived fibroblasts, we focused on motor neurons expressing G93A SOD1 mutation using two stressors: serum starvation for 72 h or $H_2O_2$ injury. NSC34, a motor neuron-like cell line expressing human G93A SOD1, compared to cells expressing the human WTSOD1, has a significant increase in cytosolic oxidative stress. SOD1 G93A cells showed an approximately twofold increase in mitochondrial ROS (Fig 2A, $P = 0.001$), which was reduced when SOD1 G93A cells were treated with P110 in a dose-dependent manner (Figs 2A and EV2A; $P = 0.003$). The ability to concentrate the TMRM probe in mitochondria was decreased in SOD1 G93A (indicative of loss mitochondrial membrane potential; $P = 0.003$) that was also improved by P110 treatment in a dose-dependent manner (Figs 2A and EV2A). A causal role for endogenous nitric oxide (NO) in motor neurons and apoptosis expressing mutant SOD1 has been reported (Lee et al, 2009). As expected, we also observed a significant increase in NO levels in SOD G93A cells (under serum starvation as an additional stressor; $P = 0.0395$), which were significantly reduced by P110 treatment; limiting mitochondrial dysfunction was sufficient to reduce NO levels from 198% in hSOD1 G93A without to 145% with P110 ($P = 0.044$; Fig EV2A). This was associated with a twofold increase in cell necrosis (measured by LDH release) in mutant SOD1 cells, which was significantly

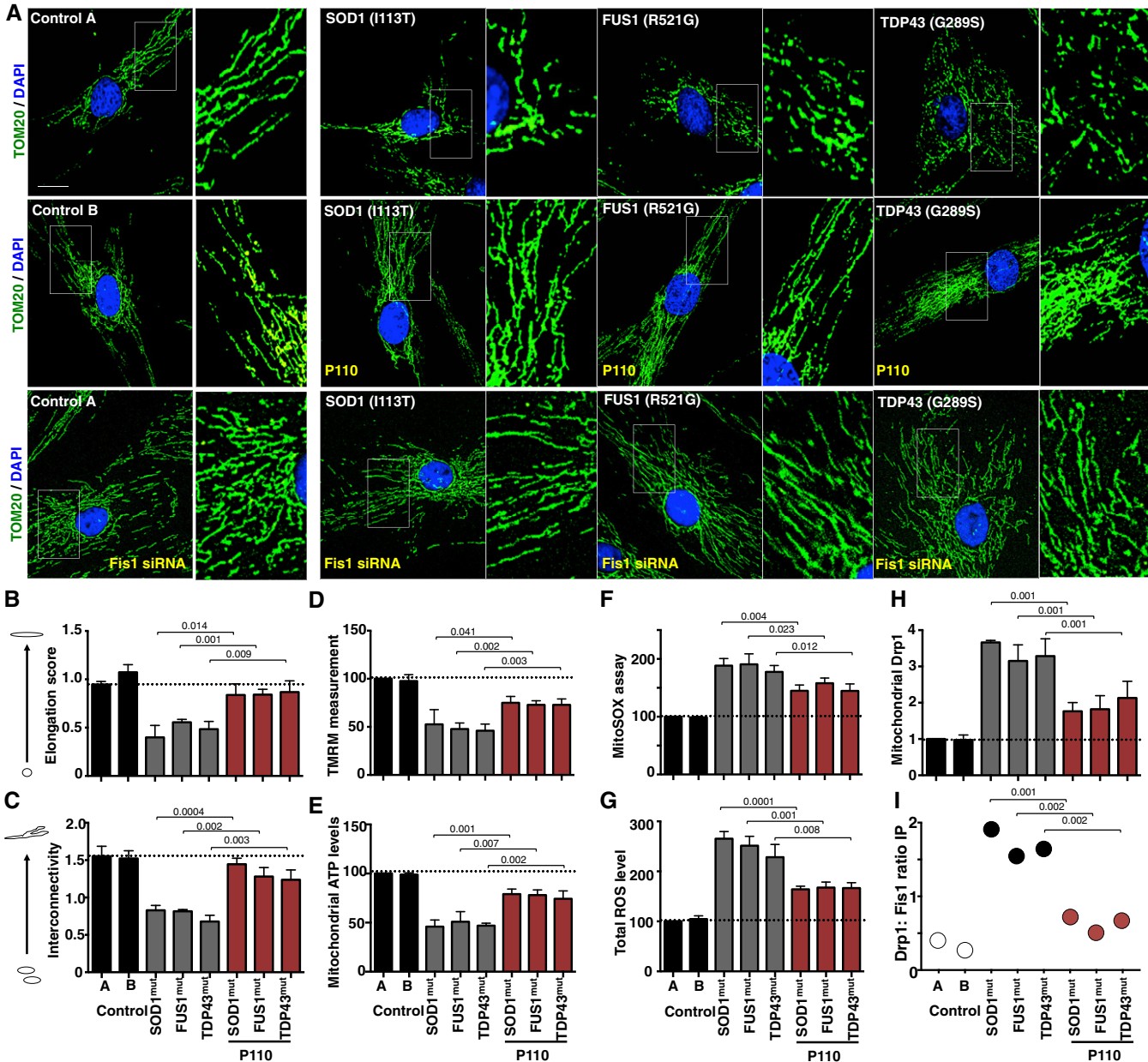

**Figure 1. Mitochondrial structural and functional defects in ALS patient-derived fibroblasts are mediated by Drp1/Fis1 interaction.**

A   Two healthy control-derived fibroblasts and three ALS patient-derived fibroblasts, each with a different genetic form of ALS, were treated with or without P110 (1 μM/24 h) for 48 h in defined medium or Fis1 siRNA and then stained with anti-TOM20 (a marker of mitochondria, 1:500 dilution). Side panels show enlarged areas of the white boxes. Scale bar: 0.5 μm.

B   Mitochondrial interconnectivity in healthy control fibroblasts and ALS patient-derived fibroblasts from the stained images was quantified using a macro in ImageJ.

C   Mitochondrial elongation in healthy control fibroblasts and ALS patient-derived fibroblasts from the stained images was quantified using a macro in ImageJ.

D   Mitochondrial membrane potential was determined using TMRM dye after 48 h in control and ALS patient-derived fibroblasts in the presence or absence of P110 (1 μM/24 h).

E   Mitochondrial ATP levels were in control and ALS patient-derived fibroblasts in the presence or absence of P110 (1 μM/24 h).

F   Measurement of mitochondrial ROS production using MitoSOX in ALS patient-derived fibroblasts in the presence or absence of P110 (1 μM/24 h).

G   Measurement of total cellular ROS production in control and ALS patient-derived fibroblasts in the presence or absence of P110 (1 μM/24 h).

H   Levels of Drp1 were examined in mitochondrial fractions by immunoblotting; VDAC was used as a loading control. Protein levels were quantified and represented as fold change of control 1.

I   Association of Drp1 with Fis1 was analyzed following co-immunoprecipitation by Western blot analysis in pooled total lysates of patient-derived cells treated in the presence or absence of P110 (1 μM/24 h) from three independent experiments. Protein levels were quantified and presented as ratio of Drp1 to Fis1.

Data information: Mean, standard deviation, and *P*-values are shown. Results are presented as percent of control. *n* = 3 performed in (H, I) duplicate, (B, C) triplicate, or (D–G) quintuplet; probability by one-way ANOVA (with Tukey's *post hoc* test). At least 100 cells/group were analyzed while blinded to experimental conditions in panels (B and C).

reduced by P110 treatment (Fig EV2A; $P = 0.0014$). Similar effect was observed following acute $H_2O_2$ treatment instead of serum starvation (Fig EV2B).

When cultured without serum for 72 h, Drp1 association with the Fis1 and not the other adaptors (Mff, Mid49, and Mid51) increased significantly, which was reduced by P110 treatment (Fig EV2C). Association of Drp1 with mitochondria was 2.6-fold higher in NSC-34 cells expressing SOD1 G93A mutant relative to control cells ($P = 0.0008$; Figs 2B and EV2D), indicating Drp1 hyperactivation, which was significantly reduced by P110 treatment ($P = 0.0008$). P110 treatment also significantly blocked the subsequent release of cytochrome c from the mitochondria ($P = 0.0397$; Fig EV3A), reduced the accumulation of active Bax on the mitochondria ($P = 0.0065$), and improved decreased Bcl-2 levels on the mitochondria in NSC-34 cells expressing SOD1 G93A mutant vs. control cells ($P = 0.0002$). Thus, P110 treatment significantly inhibited the initiation of apoptosis in these mutant cells.

Phosphorylation of Drp1 at Ser-616 by cyclin-dependent kinase (CDK) 1/cyclin B or CDK5 promotes mitochondrial fission, whereas dephosphorylation of Drp1 at Ser-637 by calcineurin facilitates its translocation to mitochondria and subsequently increases mitochondrial fission (Liesa et al, 2009; Campello & Scorrano, 2010). Therefore, a balance between Drp1 Ser-616/Ser-637 phosphorylation ratio reflects Drp1 activity. Western blot analysis of total protein lysates showed a significant increase in Drp1 phosphorylation at Ser-616 combined with a decrease in phosphorylation at Ser-637 in NSC-34 SOD1 G93A cells (Fig 2C, $P = 0.0002$). These results indicate that Drp1 hyperactivation and phosphorylation occur in NSC-34 cells expressing SOD1 G93A and that treatment with P110 inhibits this hyperactivation ($P = 0.0012$).

The ubiquitin-proteasomal system, important for maintaining protein quality control, is also compromised in experimental models of familial ALS (Cheroni et al, 2009; Dantuma & Bott, 2014; Scotter et al, 2014), and increased levels of autophagy/mitophagy markers, LC3BII and p62, have been reported in ALS models (Soo et al, 2015; Goode et al, 2016; Oakes et al, 2017). Isolated mitochondria from NSC-34 SOD1 G93A cells also had higher levels of LC3BII

($P = 0.0005$) and p62 ($P = 0.0073$) and that P110 treatment significantly reduced stalled mitophagy (Fig 2D; $P = 0.0146$ & $P = 0.0216$, respectively). Furthermore, increased c-Jun N-terminal kinase (JNK) phosphorylation, which indicates increased cellular stress, and increased LC3BII conversion and p62-enhanced accumulation in total lysates, which demonstrate altered autophagic flux, were all significantly normalized by P110 treatment in SOD1 G93A cells (Fig EV3B).

Mutant SOD1 is degraded by the proteasome and partial inhibition of proteasome activity leads to the formation of large SOD1-containing aggregates, which is thought to contribute to neuropathology (Hyun et al, 2003). Recent report indicates that the levels of proteasomal 20S constitutive catalytic subunits are significantly reduced in the spinal cord of SOD1G93A mice at an advanced stage of the disease (Kabashi et al, 2012). Similarly, we observed a decreased chymotrypsin-like proteasomal activity in NSC-34 SOD1 G93A cells ($P = 0.0087$) that was restored by P110 treatment ($P = 0.0065$; Fig 2E).

Since activated JNK is one of the mediators of ER stress-induced apoptosis (Szegezdi et al, 2006), we also determined the levels of markers of ER stress, XBP1, and ATF6α and the extent of eIF2α phosphorylation, in SOD1G93A NSC34 cells. The levels of these ER stress markers in the SOD mutant cells, which were higher than in WT cells, were significantly reduced following P110 treatment, thus indicating a functional connection between Drp1 hyperactivation and ER stress (Figs 2F and EV3C). Other ER stress markers, GRP78 and CHOP, increased by serum starvation in G93A expressing in motor neuron-like cells as compared to the WT (Fig EV3C), were significantly normalized by P110 treatment (Fig EV3C).

### Inhibition of Drp1/Fis1 interaction improves behavioral outcomes in SOD1 G93A mice

The SOD1G93A mouse model has been used since 1994 for preclinical testing of treatments for ALS (Gurney et al, 1994). Here, SOD1G93A mice (on a mixed genetic background) were treated with

**Figure 2. Expression of SOD1 G93A mutant in NSC-34 cells induces cellular stress in a Drp1-dependent manner.**

A  Mitochondria-specific ROS levels (using indicator MitoSOX), and mitochondrial integrity (using mitochondrial membrane potential TMRM) in hSOD1-WT- and hSOD1-G93A-expressing NSC-34 differentiated cells cultured under serum starvation condition in the presence or absence of P110 (0.25, 0.5, 1, 2 μM/24 h). Results are presented as percent of MOCK (empty vector).

B  Levels of Drp1 were determined in mitochondrial fractions by immunoblotting in hSOD1-WT- and hSOD1-G93A-expressing NSC-34 differentiated cells cultured under serum starvation condition in the presence or absence of P110 (0.25, 0.5, 1 μM/24 h); VDAC, a mitochondrial membrane protein, was used as a loading control. Protein levels were quantified and presented as fold change of hSOD1-WT.

C  Levels of Drp1 phosphorylation were determined in mitochondrial fractions by immunoblotting using anti-phosphorylated-S616-Drp1 or anti-phosphorylated-S637-Drp1 antibodies in hSOD1-WT- and hSOD1-G93A-expressing NSC-34 differentiated cells under serum starvation in the presence or absence of P110 (1 μM/24 h); β-actin was used as a loading control. Protein levels were quantified and presented as fold change of hSOD1-WT.

D  Levels of Parkin and LC3BII autophagy measures were determined in mitochondrial fractions by immunoblotting in hSOD1-WT- and hSOD1-G93A-expressing NSC-34 differentiated cells as in A, cultured in the presence or absence of P110 (1 μM/24 h); VDAC was used as a loading control. Protein levels were quantified and presented as fold change of MOCK (empty vector).

E  Chymotrypsin-like activity was measured using fluorogenic substrate; Suc-LLVY-AMC to measure proteasome activity in homogenates of hSOD1-WT- and hSOD1-G93A-expressing NSC-34 differentiated cells as above in the presence or absence of P110 (1 μM/24 h). Activity levels were quantified and presented as fold change of MOCK (empty vector).

F  Levels of phosphorylated-eIF2α, XBP1, and ATF-6 (measures of ER stress) in total fractions were measured by immunoblotting in hSOD1-WT- and hSOD1-G93A-expressing NSC-34 differentiated cells, cultured as above, in the presence or absence of P110 (1 μM/24 h); β-actin was used as a loading controls. Protein levels were quantified and presented as fold change of hSOD-1 WT.

Data information: Mean, standard deviation, and P-values are shown. n = 3 performed in (B, C, D, F) duplicate or (A, E) quintuplet; probability by one-way ANOVA (with Tukey's post hoc test).
Source data are available online for this figure.

either P110 or vehicle control, using Alzet osmotic mini-pumps (delivering 3 mg/kg/day). We began treatment at the age of 90 days, when the clinical/motor symptoms began, to assess the efficacy of P110 treatment in modifying disease progression (Fig 3A). We used activity chamber to determine general activity levels, gross locomotor activity, and exploration habits in rodents (Tatem *et al*, 2014), and all the following studies were carried out by an observer

blinded to the experimental conditions. Changes in mouse locomotor behavior were determined 10 days and 24 days after the initiation of P110 treatment (Figs 3B and C, and EV4). Relative to control-treated SOD1G93A mice, P110-treated SOD1G93A mice spent significantly more time exploring the chamber as well as travelled further distance (Fig 3B lower panel vs. middle panel), as measured by ambulatory distance, ambulatory episodes and time

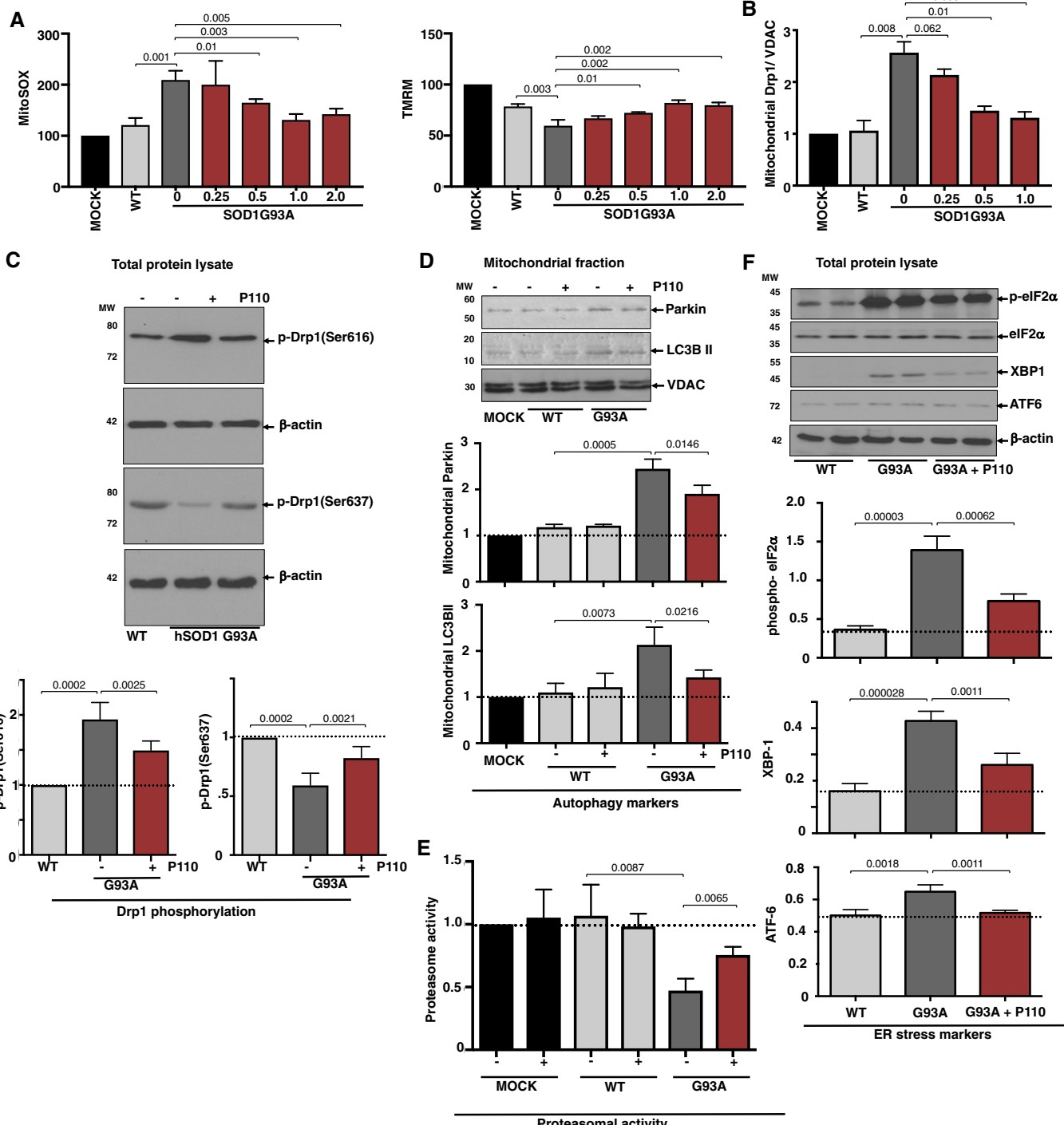

**Figure 2.**

spent exploring (Figs 3C and EV4), indicating better motor functions at both time points; their motor behavior was almost indistinguishable from that of WT mice. The exploratory behavior correlated with decreased inactive time (Figs 3C and EV4). We also observed better hind/front grip strength, a tendency toward increased jumping, a behavior associated with intense motor function, and increased locomotion (measured by center zone entries) in the P110-treated mice relative to control SOD1G93A mice, but no significant increase in body weight after 24 days of treatment (Figs 3C and EV4). Stereotypic counts are the number of times the mouse breaks the same beam in succession, without breaking an adjacent beam. We observed that P110 increases this measure, which is often associated with worsened pathology, but this could be attributed to increased movement and locomotion, as this behavior in SOD G93A mice treated with P110 was comparable to that observed in WT littermates (Fig EV4).

To determine if P110 treatment led to different behavioral outcomes, we used principal component analysis (PCA) as used before (Tanaka *et al*, 2012), to stratify the mice based on their behaviors. For each mouse, we assembled a vector containing normalized values for each behavior measured and identified the top two principal components describing the variation in mouse behavior; these accounted for 35 and 15% of the variance in the data. When we plotted the 28 mice included in the study on these principal components, we found that the WT mice segregated distinctly from the control-treated SOD1 G93A mice by silhouette score (Fig 3D), and the P110-treated mice fell between these two groups. Further analysis revealed that the first principal component (PC1) reflected general mobility of the mice: positive values for ambulatory behaviors and negative values for resting time (Fig 3D). The P110-treated mutant group showed higher values in PC1 as compared to the control-treatment mutant group, indicating improved movement and suggesting treatment benefit.

### Improved motor function is associated with improved mitochondrial structure and decreased muscle atrophy in P110-treated SOD G93A mice

Previous studies have shown that SOD1 G93A expression affects muscle structure and mitochondrial functions leading to subsequent oxidative stress and muscle atrophy (Mahoney *et al*, 2006; Dobrowolny *et al*, 2008; de Oliveira *et al*, 2014; Chen *et al*, 2015). Electron microscopy (EM) analysis of gastrocnemius muscles from WT, SOD1 G93A, and P110-treated SOD1 G93A demonstrated that fibers from SOD1 G93A transgenic mice exhibit disorganization, significant changes in mitochondrial morphology and position, and disorganization of the sarcotubular system (Fig 4A). Furthermore, mitochondria were frequently abnormally shaped ($P = 0.001$), with disturbed cristae, and some had vacuolated structures and their density per area was reduced ($P = 0.0001$; Fig 4C and D). P110-treated SOD1 G93A mice showed improved mitochondrial structure ($P = 0.004$), mitochondrial density as well as sarcotubular system while reducing the levels of altered or damaged mitochondria ($P = 0.011$; Fig 4A, C and D).

Hematoxylin and eosin staining of gastrocnemius muscle sections confirmed preserved muscle integrity in P110-treated SOD1 G93A mice. Measurement of individual muscle fiber diameter revealed a significant loss in SOD1 G93A mice as compared to WT

mice ($P = 0.00001$), which was partially corrected in P110-treated G93A mice, indicating decreased atrophy ($P = 0.0003$) (Fig 4B and E). Since previous studies have attributed muscle atrophy with increased oxidative stress, we determined 4-hydroxy-2-nonenal (4-HNE) adducts on proteins in gastrocnemius muscle (Russell *et al*, 2003; Colin *et al*, 2016). 4-HNE is a nine-carbon amphiphilic lipid-derived aldehyde formed by oxidation of $n-6$ polyunsaturated fatty acids. It easily diffuses across biological membranes and irreversibly adducts to any macromolecules. Therefore, its level in tissues correlates with ROS production (Dalleau *et al*, 2013; Liou & Storz, 2015). SOD1 G93A mice showed increased 4-HNE staining intensity ($P = 0.00001$), indicating increased oxidative stress that was significantly blunted with P110 treatment ($P = 0.0029$; Fig 4B and F).

### Inhibition of Drp1/Fis1 interaction reduces mitochondrial structural defects in motor neurons and increases survival of SOD1 G93A mice

Electron microscopy was performed to quantify the degree mitochondrial ultrastructure abnormalities in perikarya of motor neurons in the ventral horn of SOD1G93A mice. This analysis showed lower density of normal mitochondria as well as increased circular mitochondria with damaged cristae (Fig 5A) similar to previously reported observation (Milanese *et al*, 2014). Balancing the mitochondrial fission with P110 significantly reduced the number of mitochondria with damaged cristae ($P = 0.0086$). mtDNA content has been previously used to assess the mitochondrial number in various diseases, including ALS, where a significant decrease was observed in the spinal cord of patients (Wiedemann *et al*, 2002; Kim *et al*, 2010; Disatnik *et al*, 2016). We measured the levels of the transcript of the mitochondrial gene, mtND2 (mitochondria-encoded NADH dehydrogenase 2; a subunit of complex 1 located at the inner mitochondrial membrane) as a surrogate measure for mitochondrial number in the spinal cord and found a significant decrease in mtND2 levels in spinal cords from SOD1 G93A mice as compared to WT mice, which was improved by sustained P110 treatment (Fig 5A; $P = 0.0271$).

The improved overall neurological phenotype of SOD1 G93A mice in response to P110 treatment translated to an increased survival to phenotypic endpoint (Fig 5). Importantly, in this study, we started the treatment at the onset of the clinical symptoms, when the mice already showed dragging feet/knuckles (clinical score 1, CS1; Fig EV5). Median survival of vehicle-treated SOD1G93A mice was $122 \pm 2$ days, $n = 7$, whereas treatment with P110 increased the lifespan of SOD1G93A mice to $132 \pm 2$ days, $n = 14$ (Mantel–Cox test, $P = 0.007$, Fig 5B and C). Furthermore, treatment with P110 significantly delayed disease progression to terminal endpoint in these mice as assessed by the age at terminal endpoint, age at clinical score (CS3), time taken for the disease to progress from CS2 to terminal endpoint, as well as the overall increase in the duration of the disease after the onset of first symptoms and significantly improved the probability of survival post-paralysis in SOD1 G93A mice (Figs 5D–F and EV5). Moreover, sustained treatment with P110 treatment in naïve mice showed neither toxicity nor any behavioral changes after 5 months on treatment at 3 mg/kg/day (Appendix Fig S1).

Finally, to confirm that P110 treatment affected its target, we determined Drp1 association with the mitochondria in the three

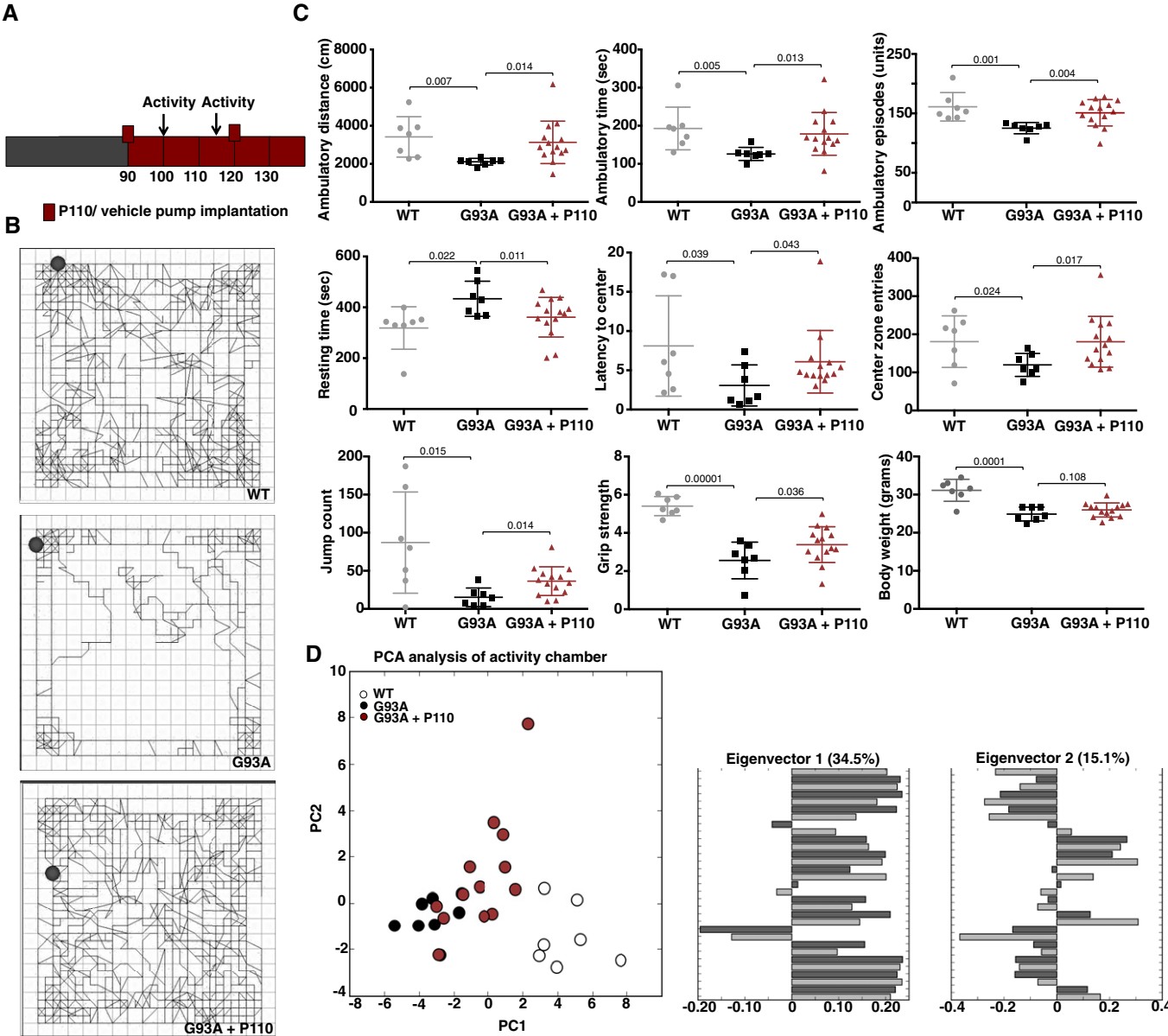

**Figure 3. Inhibition of Drp1 interaction with Fis1 using P110 improves behavioral outcomes in SOD1-G93A ALS mice.**

A   A scheme of the treatment regime beginning at the age of 90 days with pump replacement at the age of 120 days. Timing of open-field tests is also indicated.

B   Activity levels were measured in a 10-min open-field test. Example of tracks of median mice from each group: control mice, SOD1 G93A mice treated with vehicle-TAT and SOD1 G93A mice treated with P110 (3 mg/kg/day). Circles indicate starting position of the animal during the test.

C   Ambulatory distance, time, episodes, resting time, latency toward center of the chamber, number of center zone entries, and jump counts were analyzed using activity chamber after 10 days of treatment with vehicle-TAT or P110 at 3 mg/kg/day in G93A SOD1 mice. Grip strength test was carried out to assess the on muscular degeneration after 24 days of treatment.

D   PCA of the entire behavioral data shows behavioral separation between the three groups after treatment with vehicle-TAT or P110 at 3 mg/kg/day in G93A SOD1 mice. Eigenvectors 1 & 2 represent the values principal components (PC) 1 and 2 for all the behavioral endpoints analyzed.

Data information: An experimenter who was blind to genotypes and drug groups conducted all the behavior and survival studies. (C) Mean, standard deviation, and *P*-values are shown. (C and D) *n* = 7 for WT mice; *n* = 7 for G93A ALS mice + TAT; *n* = 14 for G93A ALS mice + P110; probability by one-way ANOVA (with uncorrected Fisher's LSD *post hoc* test).

groups of mice. When we analyzed the spinal cord mitochondrial fractions after 24 days of treatment (mice age was 114 days), we observed about eightfold increase mitochondrial association of Drp1 in SOD1 G93A mice as compared with WT mice ($P = 0.00012$), which was significantly corrected (approximately 65% lower)

following sustained P110 treatment (Fig 5G; $P = 0.0017$). Together, these results indicate that inhibiting Drp1 hyperactivation and increased association with the mitochondria by P110 treatment that began after disease onset reduced ALS-related pathologies in this mouse model.

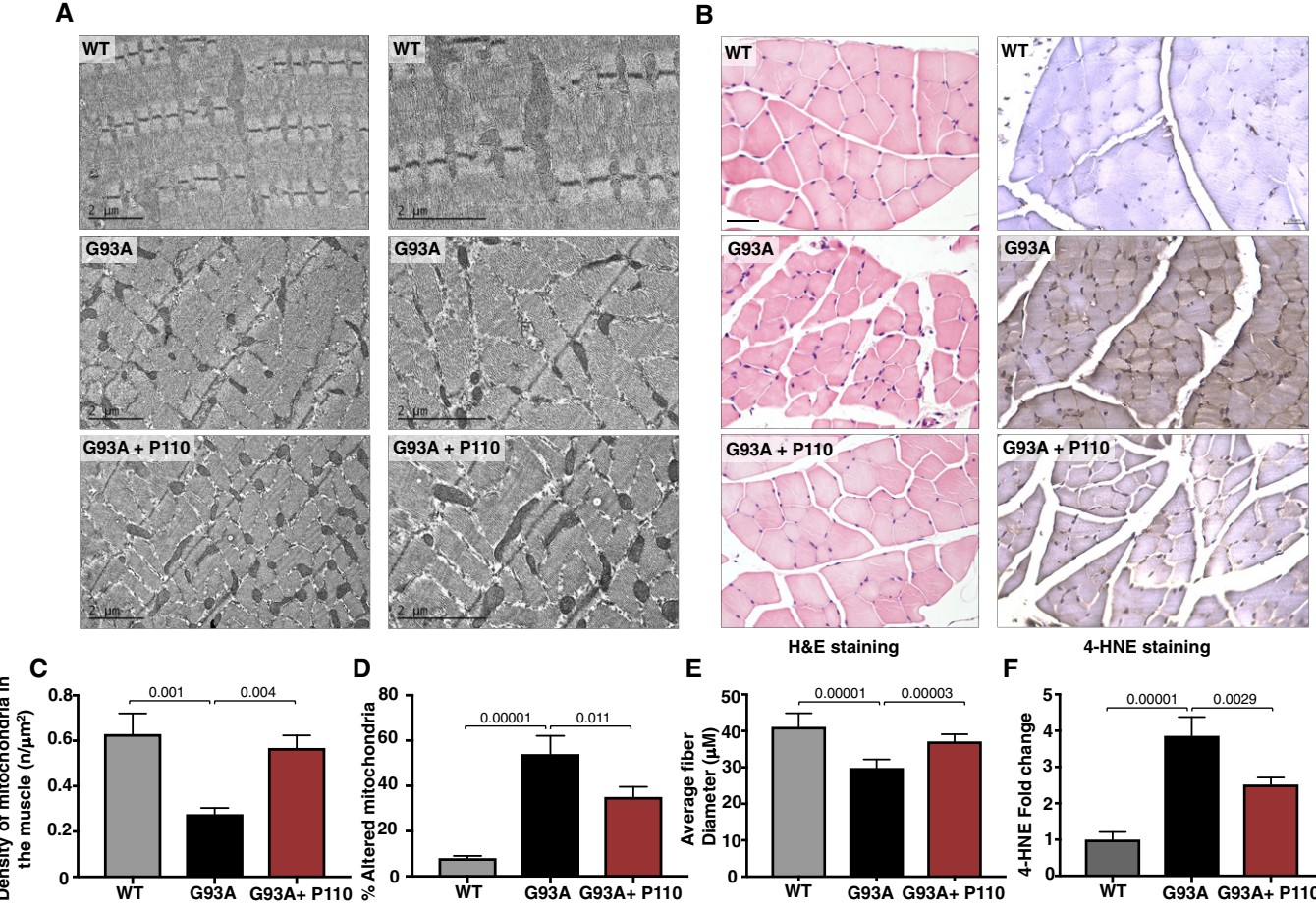

**Figure 4.  P110 treatment improves muscle and mitochondrial structure and organization, while reducing oxidative stress and atrophy.**

A   Mitochondria morphology and organization were examined in electron microscope micrographs of gastrocnemius muscle sections of WT mice, SOD1G93A mice, and SOD1G93A mice treated with P110. Scale bar: 2 μm.

B   H&E-stained gastrocnemius muscle sections of terminal stage SOD1G93A mice with and without P110 treatment, muscle fibers presenting pathological alterations including atrophy, hypertrophy, round fibers, and fibers with central nuclei. 4-HNE-stained gastrocnemius muscle sections show increased staining in SOD1G93A mice as compared to either WT or P110-treated mice. Scale bar: 50 μm.

C   Muscle from vehicle-TAT-treated G93A mice shows several disarranged areas with a significant increase in intersarcomeric area containing a few mitochondria, these effects being greatly reversed by P110. At least 40 mitochondria were analyzed in each group by an observer blinded to the experimental conditions.

D   In line with the preservation of mitochondrial architecture, P110 significantly reduces the percentage of altered mitochondria. At least 40 mitochondria were analyzed in each group by an observer blinded to the experimental conditions.

E   Quantitation of muscle fibers diameter in gastrocnemius muscle sections.

F   Quantitation of 4-HNE in gastrocnemius muscle sections of terminal SOD1G93A mice.

Data information: Treatment with vehicle-TAT or P110 at 3 mg/kg/day using Alzet 28-day pumps with pumps replaced at 120 days. Mean, standard deviation, and P-values are shown. (C–F) $n = 5$ for WT mice; $n = 5$ for G93A ALS mice + TAT; $n = 5$ for G93A ALS mice + P110; probability by one-way ANOVA (with uncorrected Fisher's LSD *post hoc* test).

## Discussion

In this study, we demonstrated that inhibition of Drp1 hyperactivation using P110, a selective peptide inhibitor of Drp1/Fis1 interaction that we developed, reduced pathological mitochondrial fission in ALS using several ALS patient-derived fibroblasts, motor neurons expressing G93A mutation, and a SOD1G93A mouse model. Specifically, we found that (i) ALS patient-derived fibroblasts carrying SOD1, FUS1, or TDP-43 mutations all exhibit a significantly increased Drp1-recruitment to the mitochondria, increased mitochondrial fragmentation, and subsequent mitochondrial

dysfunction; and these were all greatly reduced by P110. (ii) We found that neuronal cells in culture, carrying the SOD1 G93A mutant (a model of ALS in culture), also show increased mitochondrial dysfunction and increased cell death, which was dependent on Drp1 hyperactivation and was inhibited by P110 treatment. (iii) We demonstrated that inhibition Drp1/Fis1 interaction *in vivo* using P110 was safe and reduced motor/muscle deficit and increased survival in SOD1 G93A mouse model.

Despite the urgent, unmet clinical and economic need for treatment of ALS, trials of disease-modifying drugs have produced little success. Recently, edaravone, a neuroprotective drug that has

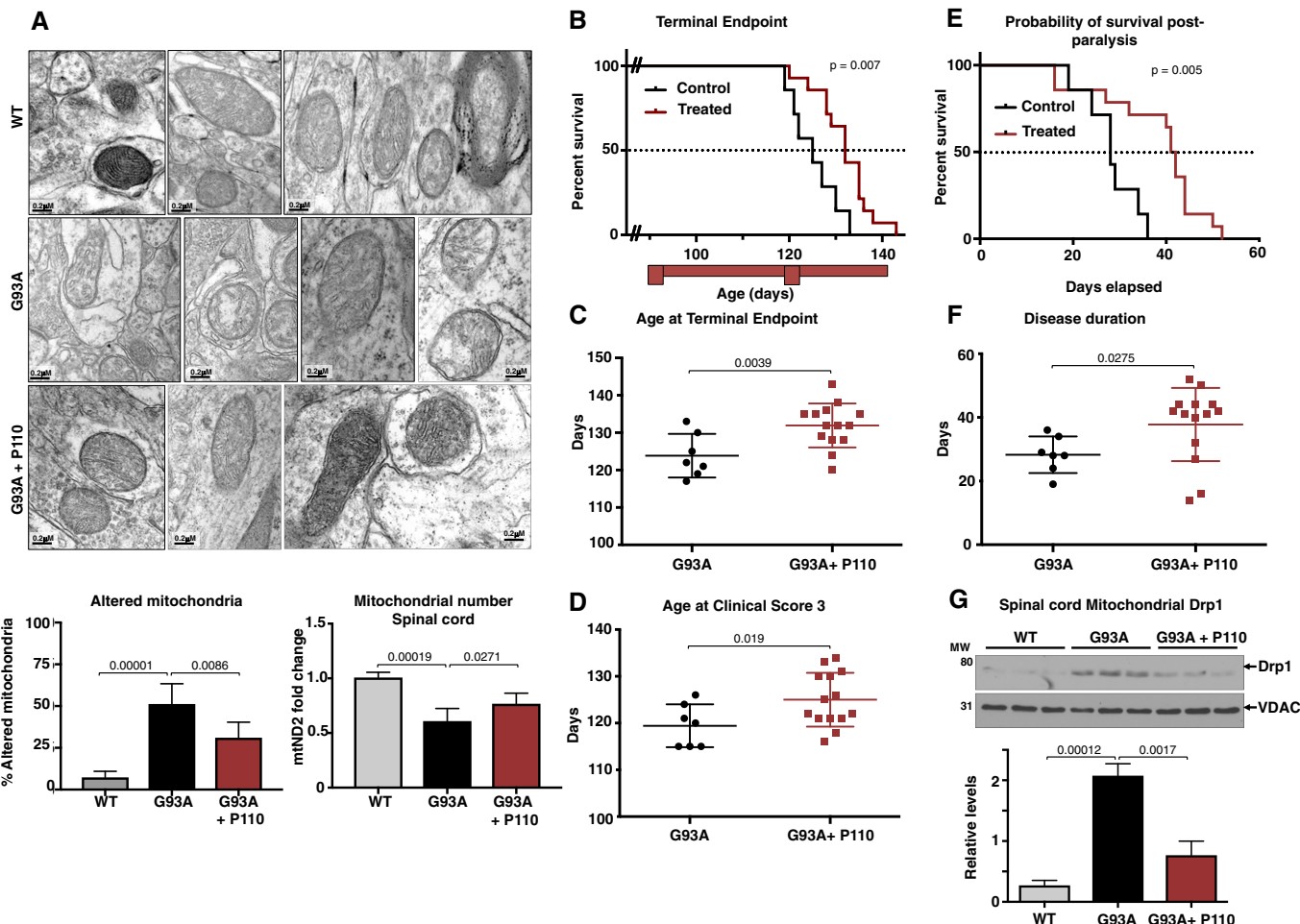

**Figure 5.  Inhibition of Drp1 association with Fis1 using P110 in the symptomatic phase improves survival and slows disease progression, in an ALS mouse model.**

A   Electron microscopy analysis of mitochondria in spinal cord from WT mice, SOD1G93A mice, and SOD1G93A mice treated with P110. Scale bar: 0.2 μm. Data are means ± SD of 20 microscopic fields, at least, from three mice per treatment group. Normal mitochondria structure was dramatically reduced and mitochondria with damaged structure increased in SOD1G93A mice. Percent mitochondria with altered structure were quantified and are represented in the graphs. At least 40 mitochondria were analyzed in each group by an observer blinded to the experimental conditions. For mitochondrial number, mtND2 levels were analyzed by real-time PCR.

B   Kaplan–Meier survival curve of G93A SOD1 (ALS) mice showing increased survival following P110 treatment (red trace) as compared to the vehicle-TAT-treated control ALS mice (black trace).

C   The terminal endpoint was significantly delayed in the P110-treated ALS mice.

D   Reaching clinical score 3 was significantly delayed in the P110-treated ALS mice.

E   Kaplan–Meier survival curve of G93A SOD1 (ALS) mice showing increased probability of survival post-paralysis following P110 treatment (red trace) as compared to the vehicle-TAT-treated control ALS mice (black trace).

F   Overall disease progression slowed down significantly, as indicated by increased days from clinical score 1 to terminal endpoint.

G   Spinal cord levels of Drp1, determined in mitochondrial fractions of WT and SOD1-G93A mice treated with either vehicle-TAT or with P110 at 3 mg/kg/day by immunoblotting; VDAC was used as a loading control. Protein levels were quantified and represented as fold change of WT (means ± SD).

Data information: Mean, standard deviation, and *P*-values are shown. (A and G) *n* = 5 for WT mice; *n* = 5 for G93A ALS mice + TAT; *n* = 5 for G93A ALS mice + P110. (B–F) *n* = 7 for G93A ALS mice + TAT; *n* = 14 for G93A ALS mice + P110. Probability by (A, C, D, F, G) one-way ANOVA (with uncorrected Fisher's LSD *post hoc* test) or (B, E) Log-rank (Mantel–Cox) test.
Source data are available online for this figure.

properties of a free radical scavenger, was the second drug to be approved by the FDA for ALS therapy (Hardiman & van den Berg, 2017; Mora, 2017; Sawada, 2017). Whereas a double-blinded placebo-controlled phase 2 study using intravenous edaravone therapy in patients with ALS failed to show a significant difference between treated and control patients, a small subset of patients did

in fact show benefit (Writing & Edaravone, 2017). Thus, there is still a large population without any disease-modifying agent (Mora, 2017).

P110 inhibitor, a 7-amino acid peptide representing a homology sequence between Drp1 and Fis1, is delivered across cell membranes and was shown previously to cross the blood–brain

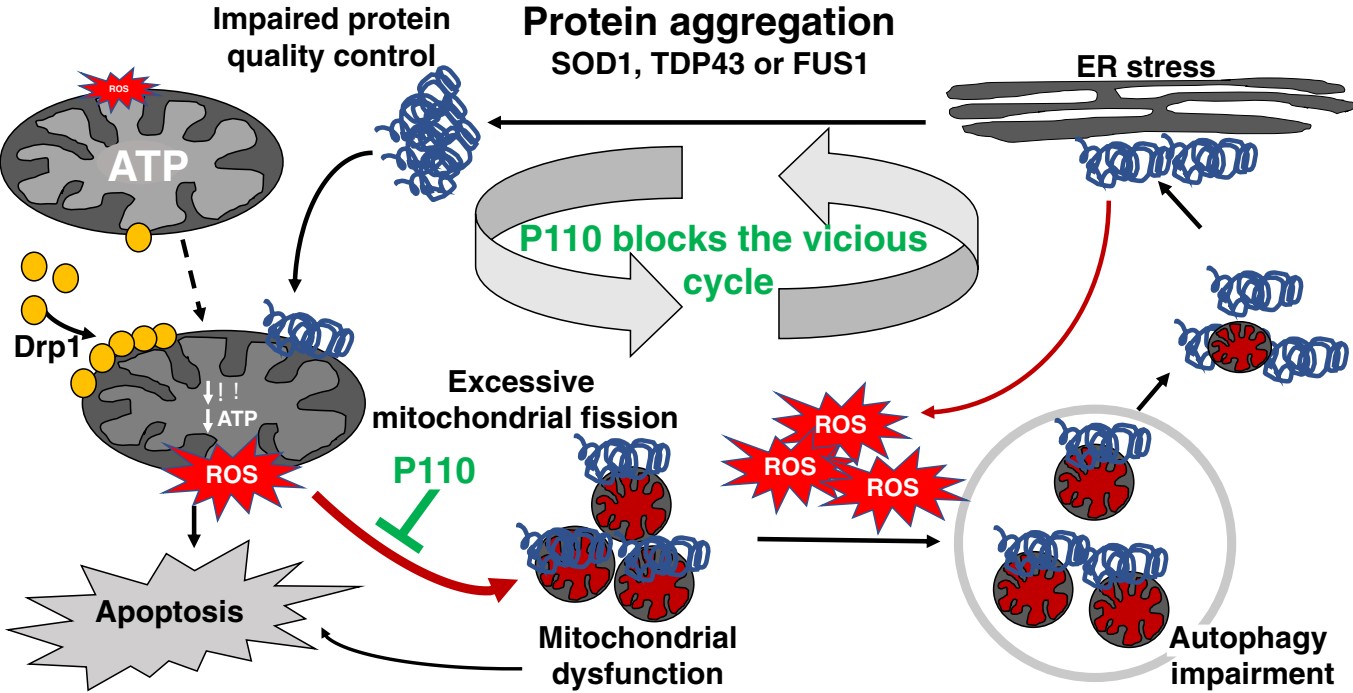

**Figure 6.   Schematic illustration of mitochondrial dynamics and the vicious cycle triggered by protein aggregation in ALS and the mechanism for the benefits of P110 treatment.**

In ALS, mutations in SOD1, TDF43, or FUS1 lead to protein aggregation which in turn starts a sequence of events ultimately culminating in metabolic and cellular failure. As a first step in the sequence of events, aggregated proteins lead to mitochondrial dysfunction causing a drop in ATP levels and concomitantly increasing ROS levels. The loss in membrane potential triggers cytochrome c release leading to the initiation of apoptotic signaling. ROS also drives increased Drp1 association with the mitochondria leading to sustained mitochondrial fission and fragmentation. Due to the drop in ATP levels, cellular functions such as autophagy and proteasomal degradation are affected leading to increased protein aggregates, which trigger ER stress. This leads to further impaired protein quality control, thus feeding back into a vicious cycle. P110, through inhibition of mitochondrial fragmentation, helps restore mitochondrial health, thereby breaking the vicious loop caused by mutated ALS-related proteins.

barrier with the TAT$_{47-57}$ carrier peptide (Guo *et al*, 2013), as do other short TAT$_{47-57}$-conjugated peptides [*e.g.*, Bright *et al* (2004), Qi *et al* (2008)]. We previously showed that P110 treatment selectively inhibits Drp1 interaction with Fis1; P110 does not affect Drp1 interaction with any other mitochondrial adaptors of Drp1 or with the mitochondrial fusion proteins OPA1, Mfn1, or Mfn2 (Qi *et al*, 2013). Importantly, P110 selectively inhibits pathological, but not physiological mitochondrial fission and fragmentation (Qi *et al*, 2013), which may explain why even a 5-month treatment of wild-type mice causes no adverse effects (Appendix Fig S1).

Because the genetic and/or environmental causes of ALS for the majority of the patients are not known, our findings of a benefit by P110 treatment in three forms of familial ALS are encouraging. Also encouraging are our findings of treatment benefits in symptomatic mice, as most patients are diagnosed at the symptomatic stage of the disease. The progression of disease in the SOD1G93A mouse model is rather rapid, resulting in many groups performing therapeutic pilot studies in the SOD1G93A mouse model beginning at the presymptomatic stage between 30 and 60 days of age, long before clinical symptoms are observed (Turner & Talbot, 2008; Eisen *et al*, 2014). Since in most cases, ALS is a sporadic disease, treatment at presymptomatic stage is clinically less relevant. Furthermore, P110 treatment significantly improved survival after the onset of the symptoms to 37 ± 5 days compared to vehicle 28 ± 5 days, which

compare favorably with the data previously published using edaravone (Ito *et al*, 2008).

Mitochondria are the primary site of ATP production, regulating calcium homeostasis and signaling, and mediating apoptosis. Therefore, mitochondrial structural and functional integrity is critical in governing cellular health. Electron microscopic studies on post-mortem tissues of ALS patients identified structural and morphological abnormalities in mitochondria of skeletal muscle, liver, spinal cord neurons, and motor cortex (Tan *et al*, 2014). In the SOD1 G93A mouse model, changes in mitochondrial structure, number, and mitochondrial fission protein levels have also been documented (Liu *et al*, 2013). Similar aberrant mitochondria have also been identified in more recent models of disease, C9ORF72, FUS1, and TDP-43 (Magrane *et al*, 2014; Lopez-Gonzalez *et al*, 2016; Onesto *et al*, 2016; Uzhachenko *et al*, 2017). Mitochondria are also found to be fragmented in cell and animal models expressing ALS-associated mutant SOD1 and TDP43 (Vande Velde *et al*, 2011; Song *et al*, 2013; Pickles *et al*, 2016; Wang *et al*, 2016).

Autophagy and mitochondrial fission are intricately linked as excessive fission leads to damaged mitochondria, which lowers the available ATP pool for autophagy to occur. Recent studies have demonstrated that autophagic alteration may be involved in ALS (An *et al*, 2014; Rudnick *et al*, 2017). Accumulation of p62 plays a role in protein aggregation in multiple neurodegenerative diseases,

including ALS (Gal et al, 2007; Lee et al, 2015; Madill et al, 2017). Furthermore, p62 serves as a shuttling factor for polyubiquitinated proteins to the proteasome and p62 actually interacts with the proteasome (Seibenhener et al, 2004; Babu et al, 2005). A deficiency in autophagy may compromise the ubiquitin–proteasome system, since overabundant p62 would impair delivery of the proteasomal substrates to the proteasome. Furthermore, as shown also here, the proteasomal activity in ALS is decreased (Cheroni et al, 2009; Kabashi et al, 2012), thereby further increasing the oxidative stress.

Mdivi-1 (mitochondrial fission inhibitor-1) was originally identified as a selective inhibitor of Drp1, blocking the entire fission machinery (Cassidy-Stone et al, 2008). When applied in multiple models of neurodegeneration, including ALS, Mdivi-1 reduced mitochondrial fragmentation and consequently improved cellular health (Cassidy-Stone et al, 2008; Luo et al, 2013; Bernard-Marissal et al, 2015). However, the selectivity of Mdivi-1 for Drp1 a has recently been challenged (Bordt et al, 2017; Smith & Gallo, 2017). Furthermore, because in addition to cell survival, physiological mitochondrial fission plays important roles in cell proliferation and differentiation and maintaining mitochondrial quality, inhibition of physiological mitochondrial dynamics is detrimental. Hence, the use of a specific inhibitor of excessive/pathological mitochondrial fission only, such as P110, as a treatment for these diseases is advantageous.

But why inhibiting excessive mitochondrial fission inhibits the pathology of neurodegenerative diseases, such as ALS? We propose that P110 treatment inhibited the vicious cycle triggered by protein aggregation, elevation of ROS, and reduction of mitochondrial function and ATP production due to excessive/pathological mitochondrial fission (Fig 6). By inhibiting Drp1/Fis1 interaction (without affecting physiological fission), mitochondrial mass and integrity increase, leading to increased ATP levels that enable better cell repair and protein quality control, and lower mitochondrial-driven ROS production. We found P110-induced inhibition of many processes related to oxidative stress, including ALS-increased ER stress (Lautenschlaeger et al, 2012; Manfredi & Kawamata, 2016), activation of JNK-related stress (Lee et al, 2016); increased ROS levels (Barber et al, 2006) and macromolecule carbonylation [through 4-HNE adduction (Niebroj-Dobosz et al, 2004)], inhibition of autophagy and reduced protein quality control [through reduced proteasome degradative activity (Kabashi & Durham, 2006; Dantuma & Bott, 2014), reduced elimination of misfolded proteins (Nedelsky et al, 2008)], and increased apoptosis triggers (Fig 6). Therefore, P110-induced better functioning mitochondria resulted in healthier neurons and muscle and improved motor functions in the mice. Importantly, P110 showed benefit in our study although the treatment was initiated only when the animals showed clinical symptoms of dragging feet/knuckles. As ALS patients are diagnosed after the symptoms become quite severe (Takei et al, 2017; Woolley & Rush, 2017), we propose that an inhibitor of Drp1-hyperactivation, such as P110, may be useful in preventing or slowing the progression of ALS in humans.

# Materials and Methods

All chemicals were purchased from Sigma-Aldrich (St Louis, MO) unless stated otherwise.

**Peptide synthesis**

Drp1 peptide inhibitor P110 and control peptide TAT were synthesized by Ontores Biotechnologies (Hangzhou, China). As previously described, (Qi et al, 2013) the peptides were synthesized as one polypeptide with $TAT_{47–57}$ carrier in the following order: N-terminus–TAT–spacer (Gly-Gly)–cargo $(Drp1_{49–55})$–C-terminus. The purity of peptides was > 90% measured by RP-HPLC Chromatogram.

*Cell culture and treatments*

NSC34 cells stably expressing WT or G93A hSOD1 were a gift from Dr. Manfredi (Weill Medical College of Cornell University, USA). NSC34 were maintained in DMEM supplemented with 10% v/v FBS and 1% (v/v) penicillin/streptomycin. For differentiation, cells were plated onto poly-d-lysine-coated plates and grown in differentiation medium, containing 1:1 DMEM/Ham's F12 supplemented with 1% FBS, 1% P/S, and 1% modified Eagle's medium non-essential amino acids. ALS patient-derived fibroblasts (ALS 1: ND29509; ALS 2: ND30327; ALS 3: ND32969) and fibroblasts of control healthy individuals (H1:AG07123; H2:AG04146) were purchased from Coriell Institute, USA. All fibroblast cultures were maintained in MEM supplemented with 15% (v/v) FBS and 1% (v/v) penicillin/streptomycin at 37°C in 5% $CO_2$-95% air. NSC34 cells were treated with P110, vehicle $(TAT_{47–57})$ at a final concentration of 1 μM every 24 h in serum-free media. Similarly, for patient-derived fibroblasts, TAT or P110 peptides were added once daily for the duration of the experiment at 1 μM final concentration. All experiments were carried out in defined serum-free media. Cells with fewer than 18 passages were used in all experiments. It should be noted that fibroblasts from healthy subjects used in previous studies (Guo et al, 2013; Qi et al, 2013) all showed similar mitochondrial morphology and functions and these parameters were unaffected by P110 treatment of these control cells. Further, 5-month P110 treatment is without adverse effects, including motor functions and mitochondrial dynamics. Therefore, P110 effects are specific for pathological and not physiological fission.

*RNAi*

Oligonucleotides for siRNA were made by QIAGEN, using siRNA previously described Fis1 (Jofuku et al, 2005) and Drp1 (Taguchi et al, 2007). siRNA was transfected to healthy control and patient-derived fibroblasts using Lipofectamine® RNAiMAX Transfection Reagent (Invitrogen) as using manufacture's protocols.

**Immunofluorescence**

Cells cultured on 8-well chamber slides were washed with cold PBS, fixed in 4% formaldehyde and permeabilized with 0.1% Triton X-100. After incubation with 2% normal goat serum (to block nonspecific staining), fixed cells were incubated overnight at 4°C with TOM20 antibody (1:500) (Santa Cruz, USA). Cells were washed with PBS and incubated for 60 min with FITC-conjugated goat anti-rabbit IgG (1:500 dilution). The cells were then washed gently with PBS and counterstained with Hoechst 33342 (1:10,000 dilution, Molecular Probes) to visualize nuclei. The coverslips were mounted with SlowFade anti-fade reagent (Invitrogen), and images were

acquired using an All-in-One Fluorescence Microscope BZ-X700 (Keyence; Guo et al, 2013; Qi et al, 2013).

## Analysis of mitochondrial morphology

We analyzed mitochondrial structure in patient-derived fibroblasts using a macro in ImageJ to assess mitochondrial interconnectivity and elongation from epifluorescence micrographs of cells immunostained for mitochondria according to the published protocol (Dagda et al, 2009). Mean area/perimeter ratio was employed as an index of mitochondrial interconnectivity, with inverse circularity used as a measure of mitochondrial elongation (Wiemerslage & Lee, 2016).

## Cell and mitochondrial health assays

### Mitochondrial membrane potential
Cells were incubated with tetra-methyl-rhodamine methyl ester (TMRM, Invitrogen) in HBSS (Hank's balanced salt solution) for 30 min at 37°C, as per the manufacture's protocol, and the fluorescence was analyzed using SpectraMax M2e (Molecular devices, using excitation at 360 nm and emission at 460 nm). All data were normalized with respect to the fluorescence intensity of the control cells.

### ATP measurements
Adenosine triphosphate levels were measured by the ATP colorimetric/fluorometric assay kit (Biovision, Milpitas, CA) using the manufacturer's protocols and read by SpectraMax M2e (Molecular devices). ATP concentration was calculated and represented as a percentage of levels in the control group.

### ROS production
For cellular ROS detection, cells were incubated with 2,7 dichlorofluorescin diacetate (DCFDA) (Abcam) 100 µM for 30 min at 37°C in the dark, and fluorescence was analyzed with excitation/emission at 495/529 nm, using SpectraMax M2e (Molecular devices). Fluorescence intensity was then normalized for cell number. To determine mitochondrial ROS production, cells were treated with 5 µM MitoSOX™ Red, a mitochondrial superoxide indicator (Invitrogen) for 10 min at 37°C, according to the manufacturer's protocol, and fluorescence was analyzed with excitation/emission at 510/580 nm, using SpectraMax M2e (Molecular devices).

### Cell death
Cytotoxicity was determined using Cytotoxicity Detection Kit (Goode et al, 2016; Roche). In brief, media was collected at endpoints (in phenol red-free DMEM) to measure the percentage of released lactate dehydrogenase activity (LDH). To quantify total LDH, cells were lysed with Triton X (1% in serum-free cell culture media) overnight at 4°C; 50 µl media or lysate was transferred with 50 µl of reaction mix in a 96-well plate and incubated at RT for 30 min in the dark. Absorbance was measured at 490 nm using SpectraMax M2e (Molecular devices), and cell death is presented as percent of released LDH of total LDH.

### Isolation of mitochondria-enriched fraction and lysate preparation
Cells were washed with cold phosphate-buffered saline (PBS) at pH 7.4 and scraped off using mannitol–sucrose (MS) buffer, containing 210 mM mannitol, 70 mM sucrose, 5 mM MOPS (3-(N-morpholino) propane-sulfonic acid), 1 mM EDTA, and a protease inhibitor cocktail, pH 7.4. Spinal cords were minced and homogenized in the lysis buffer and then placed on ice for 30 min. Collected cells or tissue were disrupted 10 times by repeated aspiration through a 25-gauge needle, followed by a 30-gauge needle (10 times). The homogenates were then spun at 800 g for 10 min at 4°C (nuclear pellet), and the resulting supernatants were aliquoted and used as total lysates. A second aliquot was spun at 10,000 g for 20 min at 4°C. The pellets were washed with lysis buffer and spun at 10,000 g again for 20 min at 4°C. The final pellets, mitochondrial-rich fractions, were suspended in lysis buffer containing 1% Triton X-100 (Guo et al, 2013; Qi et al, 2013).

### Proteasomal activity
Cells were homogenized in cold buffer (20 mM Tris–HCl pH 7.5, 2 mM EDTA) and centrifuged at 15,000 g for 10 min at 4°C. Protein concentration in supernatants was determined using the BCA protein assay (Thermo Fisher Scientific). All samples were assayed in triplicate using 10 µg of freshly prepared protein extracts. Proteasomal activity was measured using the CHEMICON Proteasome Activity Assay Kit (APT280, Millipore), as described by the manufacturer. The extracts were incubated (2 h at 37°C) with a labeled substrate, LLVY-7-amino-4-methyl-coumarin, and the proteolysis activity was monitored by detection of the free fluorophore 7-amino-4-methyl-coumarin, using a SpectraMax M2e (Molecular devices) at 380/460 nm.

### Co-immunoprecipitation
To test for Drp1/adaptor interactions, cells were chemically cross-linked using dithiobis (succinimidyl propionate) (DSP; Thermo Fisher Scientific) at a final concentration of 1.0 mM in PBS. After incubation for 1 h at room temperature, cross-linking was stopped by addition of Tris HCl (pH 7.7) to a final concentration of 20 mM. The cells were lysed with a lysis buffer containing 20 mM Tris–HCl, 150 mM NaCl, 10 mM EGTA, 10% glycerol, and 1% Triton X-100 (pH 7.4), and immunoprecipitation was performed using either Fis1 or Drp1 Abs and Pierce™ Protein A/G Magnetic Beads (Thermo Fisher Scientific) for 16 h at 4°C. The samples were washed three times in lysis buffer and boiled in SDS sample buffer containing 5% 2-mercaptoethanol to cleave the cross-linking. Samples were loaded on SDS–PAGE gel, transferred on to nitrocellulose membrane, and probe with the indicated antibodies.

### Western blot analysis
Protein concentrations were determined using the Bradford assay (Thermo Fisher Scientific). Proteins were resuspended in Laemmli buffer containing 2-mercaptoethanol, loaded on SDS–PAGE, and transferred on to nitrocellulose membrane, 0.45 µm (Bio-Rad), as before (Disatnik et al, 2013; Qi et al, 2013). Membranes were probed with the indicated antibody and then visualized by ECL (0.225 mM p-coumaric acid; Sigma), 1.25 mM 3-aminophthalhydrazide (Luminol; Fluka) in 1 M Tris pH 8.5. Scanned images of the exposed X-ray film were analyzed with ImageJ to determine relative band intensity. Quantification was performed on samples from independent cultures for each condition. The antibodies used in this study are in Appendix Table S1.

### Peptide treatment in mice

All the experiments were in accordance with protocols approved by the Institutional Animal Care and Use Committee of Stanford University and were performed based on the National Institutes of Health Guide for the Care and Use of Laboratory Animals. Adult B6SJL Tg (SOD1$^{G93A}$) 1 Gur/J male mice with a high copy number of the mutant allele and their WT littermates were purchased from the Jackson laboratory (Maine) at the age of 4–6 weeks. The animals used in the P110 treatment study were implanted with a 28-day osmotic pump (Alzet) slightly posterior to the scapulae containing TAT$_{47-57}$ carrier control peptide or P110-TAT$_{47-57}$, which delivered to the mice at a rate of 3 mg/Kg/day, as described previously (Disatnik et al, 2016). The first pump was implanted at an average age of 90 days, when the animals showed clinical symptoms, with the subsequent pump implantation 30 days later, when mice were ~120 days old.

### Animal survival and behavioral studies

An experimenter who was blind to genotypes and drug groups conducted all the behavior and survival studies. The overall survival during the study period was recorded, and the remaining mice were sacrificed when they reached terminal endpoint. The following clinical scores were used in the study; 0 = normal gait; 0.5 = slight dragging of knuckles (at least 2× during circling of arena); 1 = dragging feet/knuckles; 1.5 = single leg extremely weak/limp (little to no use for walking); 2 = weakness/limpness in two hind limbs; 3 = single leg paralysis; 4 = 2 legs paralysis; 4+ = advanced paralysis or cannot right in 20 s.

### Grip strength testing

The grip strength test was used to assess motor function and control of the fore and hind paws. Mice were allowed to grab the bar(s) on the Chatillon (Largo, Florida, USA) DFIS-10 digital force gauge while being gently pulled parallel away from the bar by the tail. The maximum force prior to release of the mouse's paw from the bar was recorded. Five trials of front paw strength and subsequent three trials of all paw strength were conducted.

### Activity chamber

The activity chamber was used to determine general activity levels, gross locomotor activity, and exploration habits in rodents. Assessment took place in an open-field activity arena (Med Associates Inc., St. Albans, VT. Model ENV-515) mounted with three planes of infrared detectors, within a specially designed sound-attenuating chamber (Med Associates Inc., St. Albans, VT. MED-017M-027). The arena is 43 cm (L) × 43 cm (W) × 30 cm (H), and the sound-attenuating chamber is 74 cm (L) × 60 cm (W) × 60 cm (H). The animal was placed in the corner of the testing arena and allowed to explore the arena for 10 min while being tracked by an automated tracking system. Parameters including distance moved, time immobile, and times spent in pre-defined zones of the arena were recorded.

### Principal component analysis

For each mouse, the behavioral and bodyweight analyses from days 100 and 114 of treatment were assembled into a vector, with a total of 31 values/vector/mouse. To avoid bias from behavioral measurements with a high range, all values were normalized (by subtracting the mean of each behavior and dividing by the standard deviation).

PCA was performed using the Python scikit-learn package. Clustering of mouse treatment groups was quantified using a silhouette score, as described previously (Cunningham et al, 2017). Briefly, for each point in a treatment group, this silhouette score (s) was defined as s = (b−a)/max(a,b), where 'a' is the mean distance between the point and other points in the same group and 'b' is the mean distance between the point and other points in the next nearest group.

### Immunohistochemistry in tissue sections

Mice were sacrificed, tissue dissected, and fixed in 4% paraformaldehyde in 0.1 M phosphate buffer, pH 7.4. Tissues were then paraffin-embedded, and sections were used for immunohistochemical staining 4-HNE (1:200; Abcam) using an immunohistochemistry Select HRP/DAB kit (EMD Millipore). The images were acquired using an All-in-One Fluorescence Microscope BZ-X700 (Keyence). Skeletal muscle was examined for pathology in transverse and longitudinal planes by H&E staining.

### Electron microscopy

Spinal cord was carefully flushed using saline, and lumbar spinal cord was dissected for further analysis. Each gastrocnemius muscle from each animal was dissected and gently stretched for 10 s before being immersed in fixing solution. Both tissues were fixed in 2.5% glutaraldehyde in 0.1 mol/l cacodylate buffer, pH = 7.4. The fixed material was sectioned at the Stanford Electron Microscopy Facility. Sections were taken between 75 and 80 nm, picked up on formvar/carbon-coated 75 mesh Ni grids and stained for 20 s in 1:1 saturated uracetate (≈7.7%) in acetone followed by staining in 0.2% lead citrate for 3–4 min for contrast. For each spinal cord, sample two tissue blocks (volume of 5 mm$^3$) were cut to obtain an average of 20 grids. Each grid included at least five cells, which were analyzed along non-serial sections. Motor neurons were selected based on classic morphological features (multipolar cells with dispersed nuclear chromatin and prominent nucleoli). Mitochondrial samples were observed in a JEOL 1230 transmission electron microscope at 80 kV, and photographs were taken using a Gatan Multiscan 791 digital camera.

Muscle architecture was examined based on intersarcomeric area, defined by the space intermingled between two sarcomers as well as density of mitochondria in the muscle, defined by the number of mitochondria per surface unit as described earlier. Mitochondria were defined as altered according to criteria being validated by previous morphological studies (i) significantly decreased electron density of the matrix (dilution, vacuolization, cavitation); (ii) fragmented and ballooned cristae (intracristal swelling); (iii) partial or complete separation of the outer and inner membranes; (iv) mitochondrial swelling. Quantitative analysis of mitochondrial damage was performed independently by two investigators, who reviewed each enlarged electron microscopy image for the presence of structurally abnormal mitochondria.

## Statistics

Data are expressed as means ± SD. Statistical analysis was assessed by ANOVA. Significance in cell culture experiments was analyzed with the Tukey's *post hoc* test. The standard Mantel–Cox log-rank test was used to assess survival. Significance of changes in neurological symptoms and tissue samples was analyzed with the Fisher's

## The paper explained

### Problem

Amyotrophic lateral sclerosis (ALS), which clinically manifests by progressive muscle atrophy and paralysis, is a fatal neurodegenerative disease with patients commonly dying of respiratory failure or pneumonia 3–5 years from initial diagnosis. Currently, the glutamate release inhibitor, riluzole, and recently approved free radical scavenger edaravone are the only medications approved by the FDA for ALS. However, there remains a strong need for new disease-modifying treatment strategies. Several recent studies suggested possible defects in mitochondrial dynamics in models of ALS, regardless of the causative mutation. However, whether Drp1 hyperactivation and its specific interaction with Fis1 play a role in the pathogenesis of ALS and neurodegeneration of motor neurons and whether its inhibition can reduce ALS pathology are unknown.

### Results

Our laboratory developed a peptide inhibitor, P110, that blocks the increased Drp1 association with mitochondria by selectively inhibiting Drp1/Fis1 interaction under pathological conditions. Mitochondrial dysfunction was evident in fibroblasts of ALS patients carrying pathogenic mutations in SOD1 (I113T), in FUS1 (fused in sarcoma; R521G), or in TDP43 (TAR DNA-binding protein 43; G289S) genes and was associated with an increased Drp1 association with the mitochondria. Further, expression of SOD1 G93A in motor neurons affected a range of signaling pathways. Correcting mitochondrial dysfunction by inhibition of pathological fission induced by Drp1/Fis1 interaction using P110 had a beneficial effect. Finally, therapeutic administration of P110 suppressed muscle atrophy and mitochondrial structural defects and enhanced motor activity and life span in SOD1-G93A ALS mouse model.

### Impact

This study establishes critical role for Drp1 hyperactivation and its interaction with a single mitochondrial outer membrane adaptor, Fis1, as a driving force behind mitochondrial dysfunction in ALS. Specific Drp1/Fis1 inhibitors, such as P110, that help restore mitochondrial dynamics without affecting the basal mitochondrial fission, may provide novel disease-modifying treatment approach. Further, this therapeutic approach might be useful in other types of muscular dystrophy with dysfunctional or aberrant mitochondrial dynamics.

LSD *post hoc* test. All analyses were conducted with GraphPad Prism software. In animal studies, we used $n = 7$–14 mice/group for behavioral tests and $n = 5$ mice/group for biochemical analysis from the same litter; the five mice included in the analysis were selected at random. For the cell culture studies, we performed at least three independent experiments, in duplicates. An observer who was blind to the experimental groups conducted all the animal studies. From the age-matched mice, one of eight TAT-treated and two of sixteen P110-treated mice were excluded from the study due to death during the surgery to implant the second pump. The data from these three mice were not included in any of the behavioral analysis.

**Expanded View** for this article is available online.

## Acknowledgments

The authors thank Mr. Juan Harrison (Takeda Pharmaceuticals) for his advice and encouragement, Dr. Andrew Evans and Michelle Halpain from the Stanford Behavioral Neuroscience Facility for the collection of mouse behavioral data, and John Perrino for technical support with EM. This work was supported, in part, by The Stanford Innovation Fund, and Takeda Pharmaceuticals' Science Frontier Fund to DM-R. Note that although Takeda sponsored, in part, the cost of this project, it was designed and executed entirely at Stanford University, and Takeda has no ownership on the findings. The animal behavioral study was supported in part by a NINDS center core grant (2 P30 NS069375 06).

## Author contributions

AUJ conducted all experiments and wrote and revised the manuscript. DM-R helped design and supervise the studies and revised the manuscript. NLS and MS performed the pump implantation and animal behavior study. ADC performed the PCA analysis on the behavioral data. HV performed the pathological assessment of tissues.

## Conflict of interest

Patents on P110 and its utility in HD, ALS and other neurodegenerative diseases have been filed by DM-R and AUJ, and P110 was recently licensed to Mitoconix Bioscience, a company that DM-R founded and serves on its board, that develops new treatment for Huntington's disease. However, none of the work in her laboratory was carried out in collaboration with or with financial support from the company. AUJ and MS both advised the company, as part of technology transfer to the company, on their work related to Huntington's disease. The other authors declare that they have no conflict of interest.

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
