## [Review Process File · EMBO Molecular Medicine]

Inhibition of Drp1/Fis1 interaction slows progression of Amyotrophic lateral sclerosis

Amit U. Joshi, Nay L. Saw, Hannes Vogel, Anna. D. Cunningham, Mehrdad Shamloo, Daria Mochly-Rosen

Review timeline:

Submission date:	19 June 2017
Editorial Decision:	06 August 2017
Revision received:	17 October 2017
Editorial Decision:	10 November 2017
Revision received:	11 December 2017
Accepted:	14 December 2017

Editors: Roberto Buccione and Céline Carret

Transaction Report:

1st Editorial Decision

06 August 2017

Thank you for the submission of your manuscript to EMBO Molecular Medicine. We are sorry that it has taken longer than we would have liked to get back to you on your manuscript.

As you will see, while recognizing the interest of this manuscript (although reviewer 2 is less convinced overall), the reviewers raise many serious, partially overlapping concerns.

The main shared issue is that of the lack of important controls and insufficient or unclear support for the core claims. These include 1) the need for solid proof that indeed inhibition of the Drp1-Fis1 interaction is occurring and underlies the effect of the peptide, 2) better explain the generally unclear MoA of the peptide, 3) lack of a peptide dose-response, 4) non-optimal histology, 5) lack of proof that the peptide crosses the BBB and many others. Please note that when deciding on whether to send your manuscript out for peer-review, I had obtained external advice from an expert advisor who did raise many of the same issues.

Finally, there is also a concern on the extent of the clinical effect of the peptide from reviewer 2 which however reviewer 1 does not appear to agree upon.

Finally, reviewer 2 also raises a significant concern on overall news value.

After our reviewer cross-commenting exercise and in depth internal discussion we agreed that the manuscript lacks the conclusiveness and strong support for the claims that would be required for publication in EMBO Molecular Medicine.

In conclusion, while publication of the paper cannot be considered at this stage, we would be willing to consider a substantially revised submission, with the understanding that the Reviewers' concerns must be addressed in full. However, we will not be asking you to undertake the BBB experiments, although your response will be required on the issue.

Since the required revision in this case appears to require a significant amount of time, additional

work and experimentation and might be technically challenging, I would therefore understand if you chose to rather seek publication elsewhere at this stage. Should you do so, and we hope not, we would welcome a message to this effect. Please note that it is EMBO Molecular Medicine policy to allow a single round of revision only and that, therefore, acceptance or rejection of the manuscript will depend on the completeness of your responses included in the next, final version of the manuscript.

***** Reviewer's comments *****

Referee #1 (Remarks):

The manuscript by Joshi and colleagues investigates a novel mechanism for the potential treatment of Amyotrophic Lateral Sclerosis (ALS) through pharmacological inhibition of Drp1 translocation to mitochondria. The authors used ALS patient-derived cellular models as well as SOD1-mutant NSC-34 motor neurons and employed a plethora of in vitro assays to determine effects of a cell-permeable peptide inhibitor P110 on various mitochondrial parameters including morphology, membrane potential, ATP levels as well as Drp1 translocation and phosphorylation. Furthermore, inhibition of Drp1/Fis1 interaction by P110 treatment was assessed on ultrastructural deficits and neurological parameters in vivo, using the ALS preclinical mouse model SOD1G93A. The investigators observed robust effects of the peptide inhibitor in both the cellular models and SOD1-mutant mice.

This is a very interesting study with robust data and solid experimental work. The manuscript is well-written and the employed methods and statistics are adequately applied. The following points need to be addressed in a revised version.

Major points:

1. The observed reversal of pathological phenotypes through P110 treatment in both in vitro and in vivo models is striking. Mechanistically, however, it remains unclear how P110 corrects all the deficits. The authors speculate that ROS might be the underlying cause for ALS phenotypes and suggest improvement through P110. Is there additional evidence that P110 acts purely as an antioxidant?
2. Figure 1: Data obtained with patient-derived fibroblasts need to be verified with an independent method, e.g. immunohistochemical staining of endogenous Drp1 mitochondrial translocation.
3. Along this line, to confirm a direct role of Drp1 translocation in ALS pathogenesis, ectopic expression of dominant-negative Drp1-K38A should be assessed in patient cells to demonstrate the proof-of-principle.
4. Has a control peptide been used in both in vitro and in vivo experiments?
5. A single concentration of the P110 peptide was used throughout the study, consistent with previously published work (PMIDs 23239023, 23813973, 24231356). A peptide titration experiment is recommended to demonstrate potency. Has the peptide ever been titrated in any of the earlier studies?

Minor points:

1. The manuscript lacks page numbers.
2. Figures 1A-G: Information on the number of counted cells should be included in the figure legend.
3. Figure 2A: A scale bar needs to be added here.
4. Figure 2C: The bar charts need to be adjusted in size according to the rest of the figure.
5. Figure 4A: The magnified areas need to be indicated in the left panel and require proper scale bars.
6. Figure 5A: Scale bars need to be added to images in the right panel.

Referee #2 (Remarks):

Joshi have contributed a manuscript on the potential role of mitochondrial fission dynamics in the evolution of amyotrophic lateral sclerosis (ALS). The work centers on the use of a putative peptide inhibitor (called P110) of Drp1-Fis1 interaction. Using familial ALS fibroblasts and a mouse immortalized embryonic spinal cord cell line they show increased fragmentation and dysfunction of mitochondria. P110 treatment mitigated these pathological changes in cell culture. They next used an in vivo transgenic mouse model of ALS that expresses mutant SOD1. They show that treatment of mutant mice with P110 beginning at the onset of disease produced a clinical improvement and extended survival. The authors conclude that Drp1 hyperactive may be a target for treatment in ALS.

This work has important strengths. It is logically designed in general, beginning with experiments in cell culture and then in vivo. Not much work has been done specifically using this P110 peptide inhibitor of mitochondrial fission. The mitochondrial imaging in the human fibroblasts is magnificent. The behavior outcome measurements in mice are pretty extensive.

This work has some important weaknesses.

The general weaknesses are:

- 1) As a general concept, novelty is lacking. Other work has been done using putative mitochondrial fission inhibitors in models of ALS, such as Mdiv-1 (Luo et al., 2013). The authors need to clarify what makes this work important and new compared to the work on Mdiv-1 in ALS models.
- 2) The effects of P110 on clinical outcomes in SOD1 mice are modest.
- 3) The in vivo experiments are fraught with difficulties.
- 4) Peptide dose response data for the in vivo experiments is needed.
- 5) The theme and in fact title of the paper centers on "inhibition of Drp1-Fis1 interaction," but where is the data in these particular ALS-related experimental systems that show the Drp1-Fis1 interactions are inhibited? Co-immunoprecipitation, FRET, or proximity ligation assay data would be useful here.
- 6) The in vivo histology work is suboptimal and imprecise.

Specific weaknesses

- 6) Figure 1. It is very strange that all of the familial ALS mutants are affecting fibroblast mitochondria equally. What is the precedent for a mitochondrial role of TDP43 and FUS in ALS?
- 7) It is very important to show that P110 peptide is targeting mitochondria in a cell culture application.
- 8) A dose-response characterization is needed for the cell culture experiments.
- 9) Identifying cultured NSC34 cells as motor neurons is inaccurate. With care and time, this line can be differentiated into motor neuron-like cells, but their characterization needs to be shown if the authors want to identify these cells as motor neurons.
- 10) Figure 2C. Is the Drp1 blot a re-probe of the pDrp1 ser616 or ser637 blot or neither?
- 11) The detail of the design for the drug administration for the in vivo experiment is lacking. They used osmotic pumps. Importantly, where were the pumps implanted: peripherally or centrally? The authors reference Disatnik et al but systemic ip injection was used in this prior work.
- 12) The authors need to show blood brain barrier penetration of P110 and very importantly brain tissue concentration of P110. This bio-distribution work needs to be done in the context of a dose-response experimental design in vivo.
- 13) Figure 4A. The skeletal muscle EM quality is poor.
- 14) Figure 4B. The paraffin histology H&E is also suboptimal, particularly for the G93A untreated image.
- 15) Figure 4C. It is not evident what is staining with the 4-HNE staining. What compartment is stained and how was this staining quantified?
- 16) Figure 5A. The EM of spinal cord needs some work. Where is this in spinal cord? They should show motor neuron mitochondria. The low magnification image of the G93A spinal cord shows mitochondria that look pretty good. The size measurements are contrary to other papers showing that mitochondria in the G93A mice swell tremendously. How was cristae damage defined and quantified?
- 17) The clinical effects of P110 are just not that robust to get excited about.

Referee #3 (Remarks):

The models appear to be sound and this is an important disease. Furthermore the idea that mitochondrial fragmentation might drive autophagy and mitophagy to excess is interesting. However, the overall message needs to be more accessible.

Major comments

1. I am not clear about the proposed mechanism. Would a diagram help? The authors mention "stalled autophagy" but do not supply any specific references regarding this (to me new) concept. Is ALS another example of impaired autophagy (Cullup et al., 2013)? This seems unlikely given the role of mitochondrial fragmentation.

2. That excessive mitochondrial fragmentation might drive mitophagy is an interesting idea, previous examples of this should be quoted. Mitochondrial fragmentation appears to be excessive in OPA1 mutants and this is linked with increased mitophagy (as in this paper) and mtDNA depletion (Elachouri et al., 2011) (Liao et al., 2017). If mitophagy is increased in ALS, what makes it excessive? Does it cause a deficiency or impair the effectiveness of mitophagy? Was mtDNA quantified? Or is it stalled mitophagy that is the problem (more explanation needed please).

Minor points

They repeatedly refer to EV1A but expanded view 1 has no "A" or indeed labels of the mutants loaded.

I am not familiar with 4-HNE staining of skeletal muscle. The reference they quote (Niebroj Dobosz) looks like CNS not muscle, where is the reference showing that this method is validated for quantitation of 4-HNE in muscle?

Cullup, T., et al 2013. Recessive mutations in EPG5 cause Vici syndrome, a multisystem disorder with defective autophagy. *Nat Genet* 45, 83-87.

Elachouri, et al., 2011. OPA1 links human mitochondrial genome maintenance to mtDNA replication and distribution. *Genome Res* 21, 12-20.

Liao, C. et al 2017. Dysregulated mitophagy and mitochondrial organisation in optic atrophy due to OPA1 mutations. *Neurology* 88, 131-142.

1st Revision - authors' response

17 October 2017

Referee #1 (Remarks):

The manuscript by Joshi and colleagues investigates a novel mechanism for the potential treatment of Amyotrophic Lateral Sclerosis (ALS) through pharmacological inhibition of Drp1 translocation to mitochondria. The authors used ALS patient-derived cellular models as well as SOD1-mutant NSC-34 motor neurons and employed a plethora of in vitro assays to determine effects of a cell-permeable peptide inhibitor P110 on various mitochondrial parameters including morphology, membrane potential, ATP levels as well as Drp1 translocation and phosphorylation. Furthermore, inhibition of Drp1/Fis1 interaction by P110 treatment was assessed on ultrastructural deficits and neurological parameters in vivo, using the ALS preclinical mouse model SOD1G93A. The investigators observed robust effects of the peptide inhibitor in both the cellular models and SOD1-mutant mice.

This is a very interesting study with robust data and solid experimental work. The manuscript is well-written and the employed methods and statistics are adequately applied. The following points need to be addressed in a revised version.

R: we thank the reviewer for their support.

Major points:

1. The observed reversal of pathological phenotypes through P110 treatment in both in vitro and in vivo models is striking. Mechanistically, however, it remains unclear how P110 corrects all the

deficits. The authors speculate that ROS might be the underlying cause for ALS phenotypes and suggest improvement through P110. Is there additional evidence that P110 acts purely as an antioxidant?

R: We appreciate the reviewer comments and have added a discussion addressing this important point. P110 is not an anti-oxidant per se; it reduces the levels of mitochondrial and cellular ROS (Fig. 1F, G 2A) by preventing mitochondrial dysfunction. Improved mitochondrial function resulted in increased ATP levels (Fig. 1E, and (Guo, Disatnik et al., 2013)), which in turn corrects impaired autophagy (Fig. 2D and E) and increases proteasomal activity (Fig. 2F), thus decreasing protein aggregates – a major cause for the ALS pathology. We also find that it increases healthy mitochondrial content (revised Fig. 4A, C, D and Fig. 5A), reduces accumulation of damaging aldehyde, 4HNE (Fig. 4B, F) which results in a healthier muscle (Fig. 4E). We now provide more detailed discussion, explaining the mechanism and benefit of P110 on p.17-18, 378-393 and added a scheme (new Fig. 6).

2. Figure 1: Data obtained with patient-derived fibroblasts need to be verified with an independent method, e.g. immunohistochemical staining of endogenous Drp1 mitochondrial translocation.

R: Immunohistochemistry showing higher levels of Drp1 on the mitochondria is relatively insensitive, as the majority of Drp1 remains cytosolic. Instead we provide new data showing increased co-IP of Drp1 with Fis1 that is blocked with P110 treatment in three patient derived cells (new Fig. 1I, Fig. EV1 G) and in a cell line expressing SOD1 mutant (Fig. EV2 C).

3. Along this line, to confirm a direct role of Drp1 translocation in ALS pathogenesis, ectopic expression of dominant-negative Drp1-K38A should be assessed in patient cells to demonstrate the proof-of-principle.

R: Such an experiment was reported in our previous study (Qi, Qvit et al., 2013), and independent study experiment of my ex-postdoc, Dr. Xin Qi (Su & Qi, 2013). We now added a complementary assay using healthy and patient-derived fibroblasts, showing that knocking down target of P110 (Fis1) improves mitochondrial structure (new Fig. 1 A, Fig. EV1 A-D).

4. Has a control peptide been used in both in vitro and in vivo experiments?

R: Yes. The control peptide (TAT₄₇₋₅₇) was used in both in vitro and in vivo efficacy experiments. In However, due to funding limitations, only P110 peptide was used in the safety study.

5. A single concentration of the P110 peptide was used throughout the study, consistent with previously published work (PMIDs 23239023, 23813973, 24231356). A peptide titration experiment is recommended to demonstrate potency. Has the peptide ever been titrated in any of the earlier studies?

R: We now added new data with dose response studies in cultured SOD1 G93A cells (new Fig. 2A-C, Fig. EV2D).

Minor points:

1. The manuscript lacks page numbers.

R: Now added

2. Figures 1A-G: Information on the number of counted cells should be included in the figure legend.

R: Now added; At least 100 cells/group were counted by an observer blinded to experimental conditions.

3. Figure 2A: A scale bar needs to be added here.

R: This panel is not Fig. EV2A and a scale bar is now added.

4. Figure 2C: The bar charts need to be adjusted in size according to the rest of the figure.

R: Now modified, accordingly

5. Figure 4A: The magnified areas need to be indicated in the left panel and require proper scale bars.

R: Fig. 4A has been modified and the enlarged areas were removed. A related information (containing multiple examples) is now provided in Fig. 5A

6. Figure 5A: Scale bars need to be added to images in the right panel.

R: Fig. 5A (Now Fig. EV2A) has been modified and scale bars were added in each panel.

Referee #2 (Remarks):

Joshi have contributed a manuscript on the potential role of mitochondrial fission dynamics in the evolution of amyotrophic lateral sclerosis (ALS). The work centers on the use of a putative peptide inhibitor (called P110) of Drp1-Fis1 interaction. Using familial ALS fibroblasts and a mouse immortalized embryonic spinal cord cell line they show increased fragmentation and dysfunction of mitochondria. P110 treatment mitigated these pathological changes in cell culture. They next used an in vivo transgenic mouse model of ALS that expresses mutant SOD1. They show that treatment of mutant mice with P110 beginning at the onset of disease produced a clinical improvement and extended survival. The authors conclude that Drp1 hyperactive may be a target for treatment in ALS.

This work has important strengths. It is logically designed in general, beginning with experiments in cell culture and then in vivo. Not much work has been done specifically using this P110 peptide inhibitor of mitochondrial fission. The mitochondrial imaging in the human fibroblasts is magnificent. The behavior outcome measurements in mice are pretty extensive.

R: We thank the reviewer for their support.

This work has some important weaknesses. The general weaknesses are:

1) As a general concept, novelty is lacking. Other work has been done using putative mitochondrial fission inhibitors in models of ALS, such as Mdivi-1 (Luo et al., 2013). The authors need to clarify what makes this work important and new compared to the work on Mdivi-1 in ALS models.

R: We thank the reviewer for the comment; indeed, we should have emphasized the novel aspects of our study.

1. Mdivi-1 was identified as a general Drp1 inhibitor (Cassidy-Stone, Chipuk et al., 2008), rather than a specific inhibitor of the pathological interaction of Drp1 with Fis1. Inhibition of physiological fission through interaction of Drp1 with the other adaptors, notably Mff, should be toxic in a chronic setting; it prevents normal mitochondrial dynamics that is required to maintain their quality. In contrast, P110 selectively inhibits pathological Drp1 activity (induced by its interaction with Fis1) and does not affect physiological activity of Drp1, through its interaction with Mff, Mid49 or Mid51 (see new Fig. II, Fig. EV1 G and Fig. EV2 C). A recent study questions the selectivity of Mdivi-1 for Drp1, showing direct effect on mitochondrial Complex 1 (Bordt, Clerc et al., 2017). We now added discussion of these points (p. 17, 367-376).
2. The study by Luo et al looked only at accumulation of mutant SOD1(G93A) inside mitochondria, depolarization of mitochondrial membrane potential and abnormal mitochondrial dynamics using Mdivi-1 which was dosed for 7 days before harvesting tissue (Luo, Yi et al., 2013). However, they never looked at mitochondrial fission per se nor did they perform any survival studies. Further, no mitochondrial fission inhibitor has been tested as yet in ALS patient-derived fibroblasts. As such, our study is the first to test the effect of inhibiting mitochondrial excessive fission/ damage in a mouse model as a potential therapeutic intervention to slow down the disease progression.

2) The effects of P110 on clinical outcomes in SOD1 mice are modest.

R: Approximately 10% of ALS cases follow a familial, mostly autosomal dominant inheritance pattern (familial ALS) while the remaining 90% of cases have no clear genetic basis (sporadic ALS) and thus are often characterized as ALS only after the onset of symptoms (Smith, Shaw et al., 2017). Thus, any treatment can be initiated only when symptoms are clear. However, most of the therapeutic strategies in animal models begin during the presymptomatic phase. In contrast, we showed efficacy when treatment with P110 began after symptoms onset; P110 treatment of SOD1 mice having a clinical score 1 (*i.e.*, the mice were dragging their feet or knuckles) significantly slowed down the disease progression (Fig. 5 B-F) and significantly improved motor functions (Fig. 3). These results compare favorably with Edaravone which has been recently approved (Ito, Wate et al., 2008). Furthermore, using patient-derived cells, we showed that P110 inhibits mitochondrial structural & dysfunction of three genetic ALS forms (with mutations in SOD1, FUS1 or TDP43). Currently, there is no treatment that significantly slows down the disease. Therefore, even a modest effect will be welcomed by patients (Akimoto, Nakamura et al., 2017, Mora, 2017, Sawada, 2017). We have added this discussion to the manuscript (p. 15, 313-317, p. 16, 339-342).

The *in vivo* experiments are fraught with difficulties.

3) Peptide dose response data for the *in vivo* experiments is needed.

R: *In vivo* dose/response study is very expensive and we simply cannot afford it. A small study was recently done in Huntington's disease, showed pharmacological effects with 0.3 mg/kg/day – 10mg/kg/day, that fits our new data in ALS cell culture model (new Figs. 2 A-C, Fig EV2D). It should be pointed out also, that a dose response *in vivo* does not affect the value of our study. Our data show that the treatment is very safe (Supplementary Fig. 1) and we do not claim that P110 is the drug to be used in humans. Likely peptides or a mimetic of P110 will be more suitable.

4) The theme and in fact title of the paper centers on "inhibition of Drp1-Fis1 interaction," but where is the data in these particular ALS-related experimental systems that show the Drp1-Fis1 interactions are inhibited? Co-immunoprecipitation, FRET, or proximity ligation assay data would be useful here.

R: We thank the reviewer for bringing it up. We provide new data studying co-immunoprecipitation of Drp1 with Fis1 and showing the selective inhibition of P110 of this interaction in both patient-derived fibroblasts and in NSC-34 SOD1G93A cells (new Fig. 1I, Fig. EV1 G, Fig. EV2 C).

5) The *in vivo* histology work is suboptimal and imprecise.

R: We have obtained help from Prof. Hannes Vogel (now co-author of the study), a pathologist who has experience with muscle and spinal cord research who reviewed and revised the histology work (H&E staining as well as EM images).

Specific weaknesses

a) Figure 1. It is very strange that all of the familial ALS mutants are affecting fibroblast mitochondria equally. What is the precedent for a mitochondrial role of TDP43 and FUS in ALS?

R: While the upstream mechanism of action of TDP43, FUS, SOD as well as C9orf72 in ALS is different, but they all end up exhausting the mitochondria due to formation of protein aggregate (Carri, D'Ambrosi et al., 2017, Cozzolino & Carri, 2012, Kawamata & Manfredi, 2010), thus increasing ROS production and triggering pathological fission. A recent review article by Smith et al highlights mitochondrial defects across all known ALS causative genes (Smith et al., 2017) . We now added a discussion of this mechanism (p. 17-18, 378-393) and a new scheme (new Fig. 6).

b) It is very important to show that P110 peptide is targeting mitochondria in a cell culture application.

R: P110 peptide was designed as an inhibitor of Drp1/ Fis1 interaction and is derived from Drp1 sequence. While it is possible to do imaging experiments with FITC-labelled peptide to look at its

interaction with Drp1/ Fis1 at or near mitochondria, it is time consuming and outside the scope of our current study, it was not conducted here.

c) A dose-response characterization is needed for the cell culture experiments.

R: We agree and have added new data (Fig 2A-C, Fig EV2D), showing that Fis1-Drp1 association is increased in ALS cell model and that these are inhibited by P110 treatment in a dose-dependent manner.

d) Identifying cultured NSC34 cells as motor neurons is inaccurate. With care and time, this line can be differentiated into motor neuron-like cells, but their characterization needs to be shown if the authors want to identify these cells as motor neurons.

R: Now corrected. These are now referred to as just NSC-34 cells.

e) Figure 2C. Is the Drp1 blot a re-probe of the pDrp1 ser616 or ser637 blot or neither?

R: We have added new figure (Fig 2C) showing individual probing as compared to loading control and the histograms represent data from 6 experiments. In the previous version, Drp1 was shown as a re-probe of pDrp1 ser616 while b-actin was a reprobe of pDrp1 ser637 blot.

f) The detail of the design for the drug administration for the in vivo experiment is lacking. They used osmotic pumps. Importantly, where were the pumps implanted: peripherally or centrally? The authors reference Disatnik et al but systemic ip injection was used in this prior work.

R: Now clarified. Disatnik *et al.* 2016 refers to the study on HD biomarkers, wherein drug was delivered using osmotic pumps. Pumps were implanted slightly posterior to the scapulae (now added p. 23, 512).

g) The authors need to show blood brain barrier penetration of P110 and very importantly brain tissue concentration of P110. This bio-distribution work needs to be done in the context of a dose-response experimental design in vivo.

R: We apologize. Indeed confirming that the drug penetrates BBB was important, but as this was demonstrated in our previous publication (Guo et al., 2013), we added a discussion of these data (p. 17, 321-324). As for bio-distribution in a context of a dose response – this is a typical pharmaceutical company work, when considering taking the compound into human studies. As this is not the aim of our current study, and because these studies are excitedly expensive, it was not conducted here.

h) Figure 4A. The skeletal muscle EM quality is poor.

R: We now replaced these with new images (see new Fig. 4A) and obtained advice of an expert pathologist, Dr. Vogel (now co-author).

i) Figure 4B. The paraffin histology H&E is also suboptimal, particularly for the G93A untreated image.

R: Now replaced with new images (see new Fig. 4B), as above.

j) Figure 4C. It is not evident what is staining with the 4-HNE staining. What compartment is stained and how was this staining quantified?

R: Now clarified (p. 12, 253-261). 4HNE, a reactive aldehyde and a product of lipid oxidation, can readily diffuse across biological membranes and therefore, its presence anywhere in the tissue is a sign of high ROS-induced lipid oxidation (Liou & Storz, 2015, Majima, Nakanishi-Ueda et al., 2002).

k) Figure 5A. The EM of spinal cord needs some work. Where is this in spinal cord? They should show motor neuron mitochondria. The low magnification image of the G93A spinal cord shows

mitochondria that look pretty good. The size measurements are contrary to other papers showing that mitochondria in the G93A mice swell tremendously. How was cristae damage defined and quantified?

R: We thank the reviewer for their comments. The mitochondria are indeed abnormal, some swollen but also showing further damage. Now, we provide a more detailed information (see p. 25-26, 555-576) and replaced the images (see new Fig. 5A). The analysis was carried out by our expert pathologist.

l) The clinical effects of P110 are just not that robust to get excited about.

R: We respectfully do not agree, considering that we initiated the treatment when the mice had symptoms (relevant to most patients, that are diagnosed after disease onset) and that this mouse model is very aggressive. Note also, that we did not suggest that P110-like treatment is a cure, but rather, that this treatment will likely slow down the progression of this disease.

Referee #3 (Remarks):

The models appear to be sound and this is an important disease. Furthermore, the idea that mitochondrial fragmentation might drive autophagy and mitophagy to excess is interesting. However, the overall message needs to be more accessible.

Major comments

1. I am not clear about the proposed mechanism. Would a diagram help? The authors mention "stalled autophagy" but do not supply any specific references regarding this (to me new) concept. Is ALS another example of impaired autophagy (Cullup et al., 2013)? This seems unlikely given the role of mitochondrial fragmentation.

R: We thank the reviewer for the suggestion and have added a new scheme to explain the mechanism (see new Fig. 6).

2. That excessive mitochondrial fragmentation might drive mitophagy is an interesting idea, previous examples of this should be quoted. Mitochondrial fragmentation appears to be excessive in OPA1 mutants and this is linked with increased mitophagy (as in this paper) and mtDNA depletion (Elachouri et al., 2011) (Liao et al., 2017). If mitophagy is increased in ALS, what makes it excessive? Does it cause a deficiency or impair the effectiveness of mitophagy? Was mtDNA quantified? Or is it stalled mitophagy that is the problem (more explanation needed please).

R: These are all excellent questions. Using mtND2 gene as a surrogate marker for mitochondrial mass, we quantified the levels in the spinal cord of SOD1 G93A mice and found a significant decrease similar to observations from ALS patients (Wiedemann, Manfredi et al., 2002) which was improved with P110 treatment (Fig 5A bottom right). This suggests that in addition to increased damage to the remaining mitochondria, due to mitochondrial depletion, there's lower ATP which affects autophagy (ATP-dependent process) leading to further cell stress and eventual failure. We added a discussion (p. 17-18, 378-393) and the new scheme to explain the mechanism (see new Fig. 6), as suggested.

Minor points

They repeatedly refer to EV1A but expanded view 1 has no "A" or indeed labels of the mutants loaded.

R: Now corrected.

I am not familiar with 4-HNE staining of skeletal muscle. The reference they quote (Niebroj Dobosz) looks like CNS not muscle, where is the reference showing that this method is validated for quantitation of 4-HNE in muscle?

R: We now provided more information about 4HNE and corrected the references (p. 12, 253-261)

References:

- Akimoto M, Nakamura K, Writing Group on behalf of the Edaravone ALSSG (2017) Edaravone for treatment of early-stage ALS - Authors' reply. *Lancet Neurol* 16: 772
- Bordt EA, Clerc P, Roelofs BA, Saladino AJ, Tretter L, Adam-Vizi V, Cherok E, Khalil A, Yadava N, Ge SX, Francis TC, Kennedy NW, Picton LK, Kumar T, Uppuluri S, Miller AM, Itoh K, Karbowski M, Sesaki H, Hill RB et al. (2017) The Putative Drp1 Inhibitor mdivi-1 Is a Reversible Mitochondrial Complex I Inhibitor that Modulates Reactive Oxygen Species. *Dev Cell* 40: 583-594 e6
- Carri MT, D'Ambrosi N, Cozzolino M (2017) Pathways to mitochondrial dysfunction in ALS pathogenesis. *Biochemical and biophysical research communications* 483: 1187-1193
- Cassidy-Stone A, Chipuk JE, Ingeman E, Song C, Yoo C, Kuwana T, Kurth MJ, Shaw JT, Hinshaw JE, Green DR, Nunnari J (2008) Chemical inhibition of the mitochondrial division dynamin reveals its role in Bax/Bak-dependent mitochondrial outer membrane permeabilization. *Dev Cell* 14: 193-204
- Cozzolino M, Carri MT (2012) Mitochondrial dysfunction in ALS. *Prog Neurobiol* 97: 54-66
- Guo X, Disatnik MH, Monbureau M, Shamloo M, Mochly-Rosen D, Qi X (2013) Inhibition of mitochondrial fragmentation diminishes Huntington's disease-associated neurodegeneration. *The Journal of clinical investigation* 123: 5371-88
- Ito H, Wate R, Zhang J, Ohnishi S, Kaneko S, Ito H, Nakano S, Kusaka H (2008) Treatment with edaravone, initiated at symptom onset, slows motor decline and decreases SOD1 deposition in ALS mice. *Experimental neurology* 213: 448-55
- Kawamata H, Manfredi G (2010) Mitochondrial dysfunction and intracellular calcium dysregulation in ALS. *Mech Ageing Dev* 131: 517-26
- Liou GY, Storz P (2015) Detecting reactive oxygen species by immunohistochemistry. *Methods Mol Biol* 1292: 97-104
- Luo G, Yi J, Ma C, Xiao Y, Yi F, Yu T, Zhou J (2013) Defective mitochondrial dynamics is an early event in skeletal muscle of an amyotrophic lateral sclerosis mouse model. *PLoS one* 8: e82112
- Majima HJ, Nakanishi-Ueda T, Ozawa T (2002) 4-hydroxy-2-nonenal (4-HNE) staining by anti-HNE antibody. *Methods Mol Biol* 196: 31-4
- Mora JS (2017) Edaravone for treatment of early-stage ALS. *Lancet Neurol* 16: 772
- Qi X, Qvit N, Su YC, Mochly-Rosen D (2013) A novel Drp1 inhibitor diminishes aberrant mitochondrial fission and neurotoxicity. *Journal of cell science* 126: 789-802
- Sawada H (2017) Clinical efficacy of edaravone for the treatment of amyotrophic lateral sclerosis. *Expert Opin Pharmacother* 18: 735-738
- Smith EF, Shaw PJ, De Vos KJ (2017) The role of mitochondria in amyotrophic lateral sclerosis. *Neuroscience letters*
- Su YC, Qi X (2013) Inhibition of excessive mitochondrial fission reduced aberrant autophagy and neuronal damage caused by LRRK2 G2019S mutation. *Human molecular genetics* 22: 4545-61
- Wiedemann FR, Manfredi G, Mawrin C, Beal MF, Schon EA (2002) Mitochondrial DNA and respiratory chain function in spinal cords of ALS patients. *Journal of neurochemistry* 80: 616-25

2nd Editorial Decision

10 November 2017

Thank you for the submission of your revised manuscript to EMBO Molecular Medicine. We have now received the enclosed reports from the referee who was asked to re-assess it. As you will see the reviewer is now globally supportive and I am pleased to inform you that we will be able to accept your manuscript pending the following final amendments:

- 1) Please address the minor change commented by referee 1. Please provide a letter INCLUDING the reviewer's reports and your detailed responses to their comments (as Word file).

***** Reviewer's comments *****

Referee #1 (Remarks for Author):

Joshi and colleagues present an overall very convincing study on a potential therapeutic approach for the treatment of ALS. All of my previous comments have now been adequately addressed in the revised version of the manuscript. I recommend acceptance of the paper pending correction of one remaining issue:

1. The new Figure EV1 G, as mentioned in the rebuttal letter, is missing from the revised manuscript and needs to be included.

2nd Revision - authors' response

11 December 2017

Response to the reviewer:

Referee #1: The new Figure EV1 G, as mentioned in the rebuttal letter, is missing from the revised manuscript and needs to be included.

R: We now have added the missing figure in the revised manuscript.

Corresponding Author Name: DARIA MOCHLY-ROSEN

Journal Submitted to: EMBO MOLECULAR MEDICINE

Manuscript Number: EMM-2017-08166-V2